# A click chemistry-mediated all-peptide cell printing hydrogel platform for diabetic wound healing

Jinjian Huang[1,7], Rong Yang[2,3,7], Jiao Jiao[4,7], Ze Li[1], Penghui Wang[2], Ye Liu[5], Sicheng Li[1], Canwen Chen[1], Zongan Li[6], Guiwen Qu[5], Kang Chen[1], Xiuwen Wu ![ORCID][1] ✉, Bo Chi ![ORCID][2] ✉ & Jianan Ren ![ORCID][1] ✉

High glucose-induced vascular endothelial injury is a major pathological factor involved in non-healing diabetic wounds. To interrupt this pathological process, we design an all-peptide printable hydrogel platform based on highly efficient and precise one-step click chemistry of thiolated γ-polyglutamic acid, glycidyl methacrylate-conjugated γ-polyglutamic acid, and thiolated arginine-glycine-aspartate sequences. Vascular endothelial growth factor 165-overexpressed human umbilical vein endothelial cells are printed using this platform, hence fabricating a living material with high cell viability and precise cell spatial distribution control. This cell-laden hydrogel platform accelerates the diabetic wound healing of rats based on the unabated vascular endothelial growth factor 165 release, which promotes angiogenesis and alleviates damages on vascular endothelial mitochondria, thereby reducing tissue hypoxia, downregulating inflammation, and facilitating extracellular matrix remodeling. Together, this study offers a promising strategy for fabricating tissue-friendly, high-efficient, and accurate 3D printed all-peptide hydrogel platform for cell delivery and self-renewable growth factor therapy.

Diabetes mellitus (DM) is a chronic metabolic disorder that affects almost 463 million adults worldwide. Its incidence is continuously growing due to increasing population age[1]. Approximately a quarter of DM patients develop diabetic foot ulcers (DFUs), with a five-year mortality rate of 30.5%, which is similar to that of cancer (31%)[2]. DFUs are associated with chronic pain, decreased self-care abilities, prolonged infections, high risk of foot amputation, and heavy economic burdens of roughly US$ 8.5 billion globally[3]. Conservative medical treatments, such as off-loading therapy, negative pressure therapy, surgical debridement, and antibiotics, often do not achieve satisfactory outcomes and are associated with adverse effects[4,5]. Therefore, efforts to develop new therapies for such complicated and chronic wounds are required.

The process of wound healing includes three overlapping and dynamic phases of inflammation, proliferation, and tissue remodeling[6]. Throughout the entire process, angiogenesis plays fundamental roles by facilitating immune cell migration, increasing nutrition and oxygen supply, and modulating scar formation[7,8]. Unlike non-diabetic wounds, the initiation of angiogenesis is impaired in diabetic wounds due to high glucose (HG)-induced endothelial cell

[1]Research Institute of General Surgery, Jinling Hospital, Affiliated Hospital of Medical School, Nanjing University, Nanjing 210093, China. [2]State Key Laboratory of Materials-Oriented Chemical Engineering, College of Biotechnology and Pharmaceutical Engineering, Nanjing Tech University, Nanjing 211816, China. [3]Key Laboratory of Medical Molecular Virology (MOE/NHC/CAMS), School of Basic Medical Sciences, Fudan University, Shanghai 200032, China. [4]Department of Rehabilitation, The First Affiliated Hospital of Nanjing Medical University, Nanjing 210029, China. [5]School of Medicine, Southeast University, Nanjing 210009, China. [6]Jiangsu Key Laboratory of 3D Printing Equipment and Manufacturing, NARI School of Electrical and Automation Engineering, Nanjing Normal University, Nanjing 210042, China. [7]These authors contributed equally: Jinjian Huang, Rong Yang, Jiao Jiao. ✉e-mail: wuxiuwen@nju.edu.cn; chibo@njtech.edu.cn; JiananR@nju.edu.cn

damage and organelle dysfunction[9–11]. The resultant poor vascularization of wounds leads to tissue hypoxia and triggers the activation of hypoxia-elicited inflammation cascade[12,13], consequently impeding the wound-healing process. Thus, the angiogenic function is a major target of new therapies for diabetic wound healing.

In recent years, growth factors, such as vascular endothelial growth factor (VEGF), have shown promising results regarding improving diabetic wound healing[14]. VEGF has different actions depending on the VEGF receptor (VEGFR) that is stimulated. VEGFR-1 mainly promotes inflammation, while VEGFR-2 contributes to angiogenesis[15]. A higher expression of VEGF is associated with improved outcomes of DFUs[16]. However, a meta-analysis on clinical trials of growth factors revealed that the efficacy of growth factors for diabetic wound healing remains questionable and varies with their category, usage, and frequency[17]. This is partially due to the inability of growth factors to stimulate wound healing because they are diluted and flushed away by wound exudation or inactivated with time due to hydrolysis by tissue proteases[18,19]. Moreover, diabetic wound care consumes longer time and is a great nursing load to appropriately care for the dressings that contain growth factors[20]. These dressings supplement the wounds with the required growth factors.

Several previous studies have reported on different growth factor delivery systems that can prolong the release time[21,22], have smart releasing profiles[23], or increase protein stability[24]. These systems were based on intelligent hydrogels, polymeric fibers, microspheres, microbubbles, diverse nanoparticles, liposomes, and other hybrid materials[25–27], which have enhanced therapeutic effects on tissue repair. However, concerns still exist regarding the dose attenuation or inactivation of growth factors during prolonged periods of diabetic wound healing since the growth factors are not self-supplemented. The use of cellular vehicles to release therapeutic agents has recently offered the prospect of biocompatibility, large-loading capacity, and long in vivo lifespan[28]. Still, to replenish diabetic wounds with continuous and self-renewable VEGF was rarely explored and challenging.

Moreover, in case of urgent needs for complicated wounds, the majority of drug carriers are not able to present customized shapes[29], thus creating gaps between bench and bedside. Hence, it is critical to develop delivery systems that can achieve shape-adaptation promptly and produce growth factors continuously for wounds to accelerate healing. Digital light processing (DLP) printing is a high-resolution fast-speed additive manufacturing technology that forms desirable 3D structures through photopolymerization reaction to solidify hydrogels[30], which has been considered a type of promising biomaterial effective for wound healing[31–34]. Some traditional peptide-based hydrogel bioinks have been developed for cell-based photocuring printing such as methacrylate gelatin[35], methacrylate silk[36], and methacrylate decellularized extracellular matrix[37]. The photocuring printability and cytocompatibility of these biomaterials are well characterized. Nevertheless, these bioinks present uncontrollable variability in their composition between batches due to the differences in sources and processing methods[38], thus making them less reproducible and stable during applications. Therefore, it is of great importance to produce a similar all-peptide bioink but with a definite chemical composition.

In this work, we design a cell-printable hydrogel platform using thiolated γ-polyglutamic acid (γ-PGA-SH), glycidyl methacrylate-conjugated γ-polyglutamic acid (γ-PGA-GMA), and thiolated arginine-glycine-aspartate (RGDC) sequences, in which RGDC is an integrin receptor that supports cell adhesion and spread (Fig. 1A, B). This platform is constructed via one-step gelation with the inherent characterizations of high efficiency and precision of thiol-ene click reaction (Fig. 1C). The resultant peptide-based hydrogel platform holds promise as efficacious cell and growth factor delivery vehicles and creates a living material that closely mimics the physicochemical properties of skin tissue. Moreover, human umbilical vein endothelial cells

(HUVECs) are genetically modified to HUVECs*vegf165+* with the lentivirus carrying the *VEGF 165* transcript for overexpression (Fig. 1D). In contrast to the traditional strategy of loading growth factors, the resultant HUVECs*vegf165+* that play the role of cellular vehicles are responsible for the successive generation of VEGF 165 when the cells are loaded into the hydrogel platform by DLP printing. Although previous studies have shown the effectiveness of VEGF in diabetic wound healing, the underlying mechanisms are not fully clear and are attributed to tissue angiogenesis. We clearly elucidate the signaling mechanism of VEGF 165, which hinders Bax-induced pore formation on mitochondrial membranes (MMs), thus improving apoptosis of endothelial cells and reserving the potential for vascularization. Then, using a diabetic rat model, the therapeutic efficacy of such a cell-laden hydrogel platform in diabetic wound healing is evaluated comprehensively (Fig. 1E), including wound closure rate, angiogenesis, epithelialization, scar risks, hair formation, stromal cell metabolism, tissue hypoxia, and inflammation.

## Results

### One-step light curing all-peptide hydrogel was designed and optimized

γ-PGA-GMA was synthesized by the epoxy ring opening reaction of γ-PGA and GMA. The synthesis of γ-PGA-GMA was confirmed by the $^1$H nuclear magnetic resonance (NMR) and Fourier transform infrared (FTIR) spectra. As shown in Fig. 2A, the spectrum of γ-PGA-GMA presented two newly emerged peaks at 6.27 ppm (a) and 5.64 ppm (b), both of which corresponded to the protons of the vinyl groups (C = C) of GMA. Comparing the integral areas of vinyl group signals at 5.64–6.27 ppm with those of the methine proton signals at 4.13–4.52 ppm (c), the substitution degree of GMA was calculated to be 28%. In the FTIR spectrum (Fig. 2B), the absorption peak at 1637 cm$^{-1}$ appeared in γ-PGA-GMA, which was attributed to the vinyl groups of GMA, thus further validating successful conjugation.

Moreover, γ-PGA-SH was prepared by a facile one-step lactamization reaction between the amino groups of cysteamine and the carboxyl groups of γ-PGA. The $^1$H NMR and FTIR spectra were used to verify the modification of thiol groups. As shown in Fig. 2A, new resonant peaks of γ-PGA-SH at 3.36 ppm (d) and 2.87 ppm (e) were clearly observed, which were assigned to the methyne (NH$_2$CHCOOH-) and terminal methylene (SHCH$_2$-) of γ-PGA-SH, respectively. The fraction of thiol groups conjugated on γ-PGA was estimated to be 46% based on the integration ratio of the methyne units at 3.25–3.56 ppm to α-protons of γ-PGA backbone at 4.13–4.52 ppm (f). Moreover, the FTIR spectrum of γ-PGA-SH revealed a new -SH stretching peak at 2596 cm$^{-1}$ (Fig. 2B), which confirmed the success of modification as well.

γ-PGA-GMA, γ-PGA-SH, RGDC, and a photoinitiator LAP were simply mixed to produce hydrogels based on thiol-ene click reaction. The click reaction refers to an approach to develop a set of fast, highly reliable, yield, and selective reactions for the rapid synthesis of useful new compounds and combinatorial libraries through heteroatom links (C-X-C)[39,40]. To examine the feasibility of gelation and optimize the physical properties of hydrogels, γ-PGA-GMA and γ-PGA-SH with the different constitutions (4.5, 5.0, 5.5, and 6.0 w/v%) were applied, which were named as Gel 1, Gel 2, Gel 3, and Gel 4 with increasing polymer concentrations (Supplementary Table 1). The hydrogel could be crosslinked quickly upon exposure to blue light (*n* = 405 nm) because the change in the storage modulus (G′) of Gel 2 was tightly coordinated to the control of light on and off (Fig. 2C). Compared with our previously reported γ-PGA-based hydrogel, addition of LAP in the pre-gel mixture significantly reduced gelation time from over 7 min to within 20 s, and saved materials[41], which made this gel formula more suitable as a bioink for photocuring printing.

Moreover, we further investigated the reactivity of RGDC upon blue light exposure of the aqueous solution by mixing RGDC (0.1 w/v%) and γ-PGA-GMA (5.0 w/v%). Following sufficient dialysis of post-

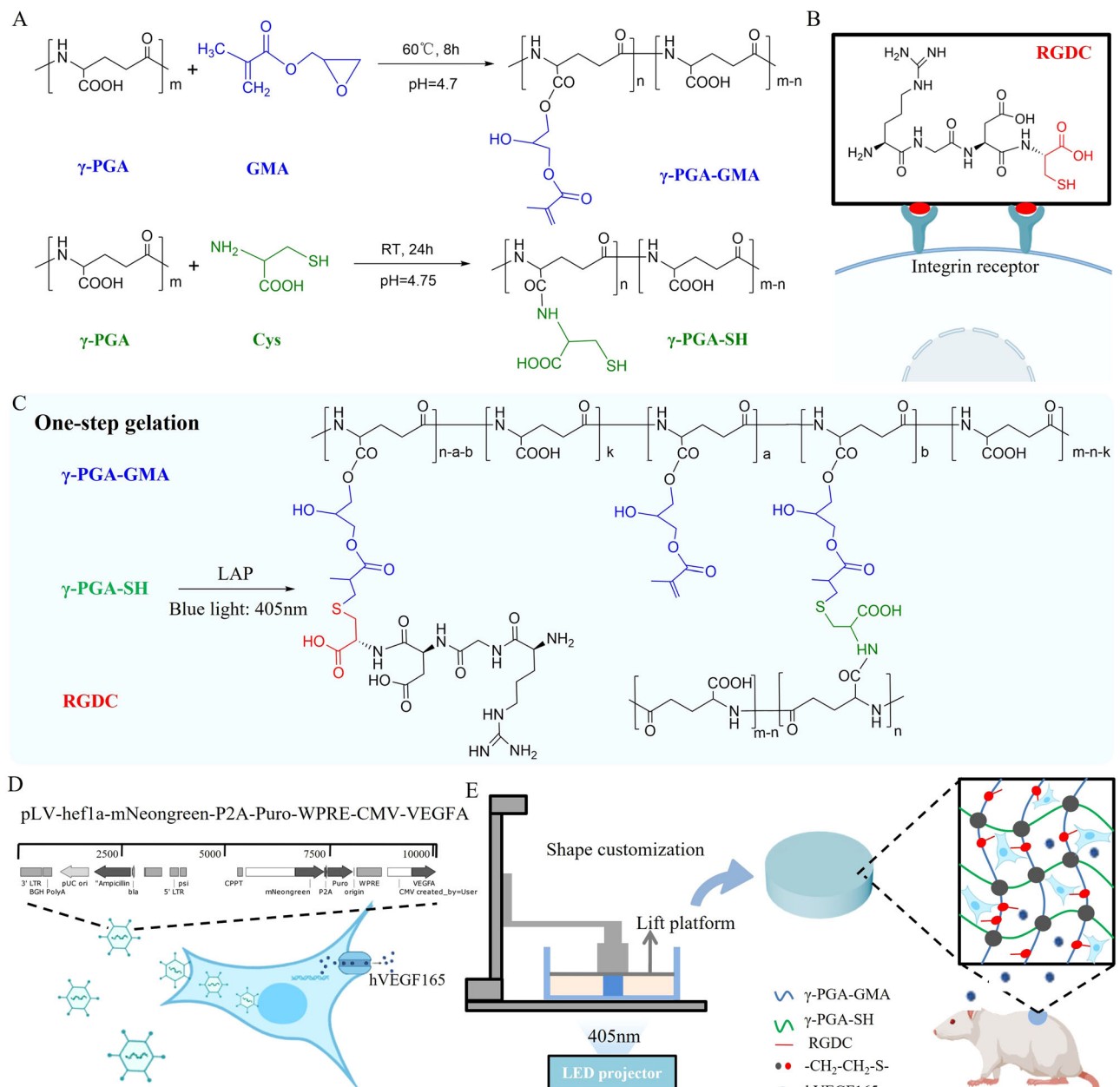

**Fig. 1 | Design of cell-printable peptide hydrogel platform to produce self-renewable VEGF 165 for diabetic wound healing. A** Synthetic routes of γ-PGA-GMA and γ-PGA-SH. RT, room temperature. **B** Chemical structure of RGDC, which presents additional thiol groups compared to RGD. **C** One-step generation of all-peptide hydrogels using thiol-ene click reaction of γ-PGA-GMA with γ-PGA-SH and reactive γ-PGA-GMA, new resonance signals in the range of 2.8–2.9 ppm (g), attributed to RGDC peptide sequences, appeared, and the peak of the vinyl groups at 5.64–6.27 ppm was reduced with almost 21% of γ-PGA-GMA modified with RGDC (Supplementary Fig. 1). The findings showed that the thiol-ene click reaction involved not only γ-PGA-GMA and γ-PGA-SH but also γ-PGA-GMA and RGDC. In other words, RGD, the important cell adhesion peptide sequence, existed in the gel system in a stable state of covalent crosslinking rather than a free state[41], which created a persistently cell-attachable and -friendly hydrogel scaffold[42].

The variation of polymer concentrations regulated the physical properties of the all-peptide hydrogel. Figure 2D, E revealed that the increase in polymer concentration decreased the pore size and improved pore density because of the enhanced crosslinking of

RGDC. LAP, a type of photoinitiator stimulated by blue light. **D** Transfection of *VEGF 165* transcript-carried lentivirus in HUVECs for overexpressing VEGF 165. **E** Printing the HUVEC*vegf165+*-laden peptide hydrogels for diabetic wound healing using sustained-release VEGF 165.

polymers. Moreover, as shown in the rheological curves (Supplementary Fig. 2), the G′ of hydrogels was increased with the increase in polymer concentration (Fig. 2F). The compressive strain-stress curves of different hydrogels are depicted in Fig. 2G. It showed that the compressive strength was gradually increased, but the strain at break was decreased due to the increase of polymer concentrations (Fig. 2H, I). Gel 3 presented the largest fracture energy, as a result of balance in compressive stress and flexibility (Fig. 2J). Moreover, the swelling ratio of Gels 3 and 4 was larger than that of Gels 1 and 2 (Supplementary Fig. 3), and their stability was also improved compared to that of Gels 1 and 2 (Supplementary Fig. 4), even though all hydrogels achieved complete hydrolysis. In the hemolysis test, the hemolysis rates of Gels 1, 2, 3 and 4 were <1% (Supplementary Fig. 5), which indicated perfect cytocompatibility[43].

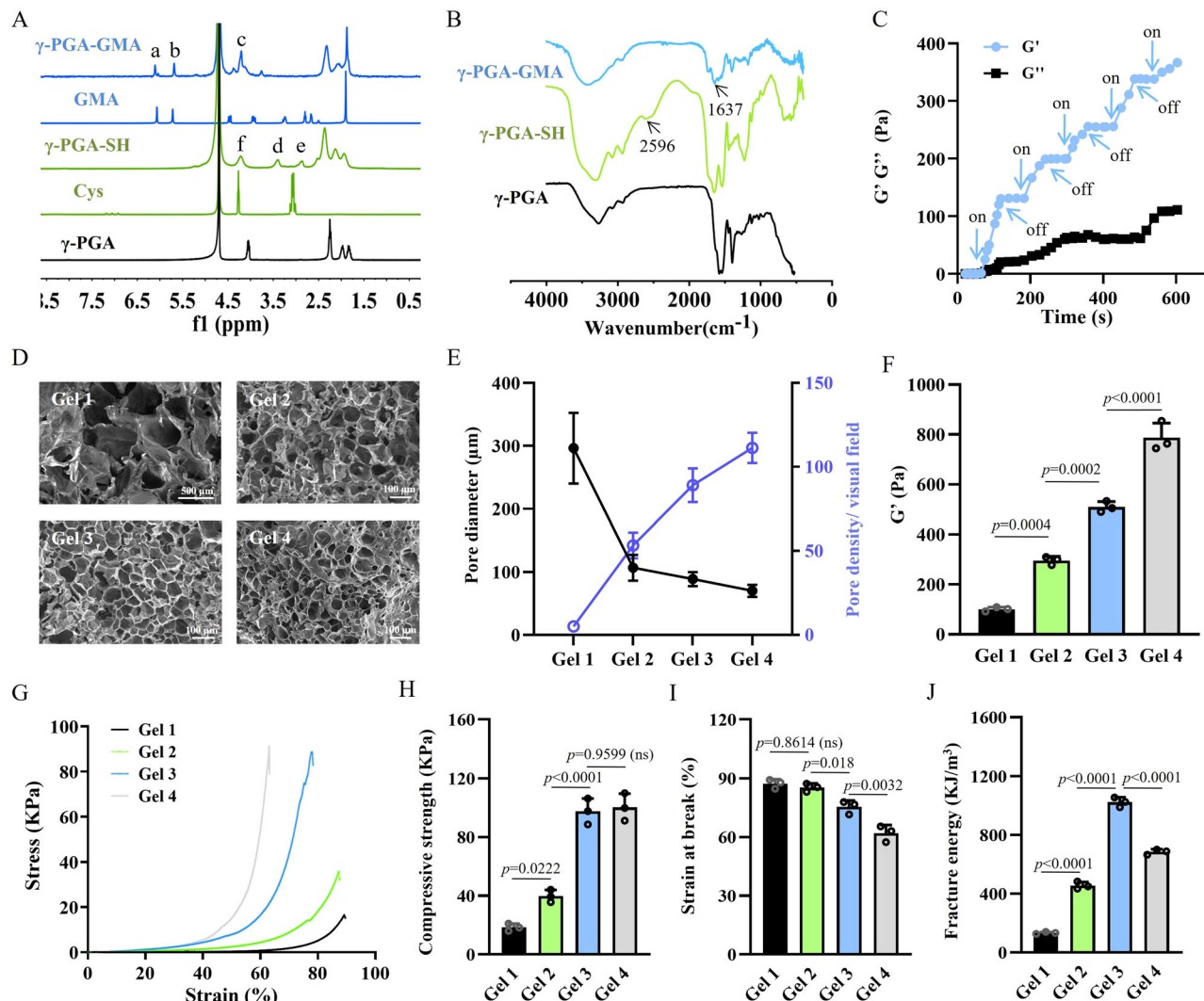

**Fig. 2 | Synthesis and optimization of the all-peptide hydrogels. A** $^1$H NMR spectrum of modified γ-PGA. a: δ = 6.27 ppm; b: δ = 5.64 ppm; c: δ = 4.13–4.52 ppm; d: δ = 3.36 ppm; e: δ = 2.87 ppm; f: δ = 4.13–4.52 ppm. **B** FTIR spectrum of modified γ-PGA. **C** Time-sweep rheological tests of the hydrogels with five cycles of blue light on and off. **D** Representative images of hydrogel micromorphology based on three repeated experiments with similar results. **E** Morphological analysis revealed the reduction in pore size and increase in pore density with increase in polymer concentration. $n = 3$ independent experiments. **F** Rheological results showing an increase in G′ of hydrogels when the polymer concentration is increased. $n = 3$ independent experiments. **G** Representative compressive stress-strain curves of different hydrogels. **H** Compressive strength of hydrogels with relatively high polymer concentration (Gel 3 and Gel 4) is increased compared to that of hydrogels with lower polymer concentration (Gel 1 and Gel 2). $n = 3$ independent experiments. **I** Strain at break of hydrogels is decreased with increase in polymer concentration. $n = 3$ independent experiments. **J** Fracture energy of Gel 3 is the highest among all the groups of hydrogels. $n = 3$ independent experiments. The $p$ values in the figures (**F**, **H–J**) are determined by one-way ANOVA followed by Tukey's multiple comparisons test. Data are presented as mean ± SD. ns not significant. Source data are provided as a Source Data file.

## High-efficiency and accurate DLP printing platform was generated based on the all-peptide hydrogel

The formula of Gel 3, which presented optimal mechanical properties, was chosen for the evaluation of DLP printability. DLP printing is an advanced light-curing printing technology that can trigger localized photo-polymerization using a digital projection and enables fast and precise printing of 3D hydrogel structures[44,45]. The light exposure time and layer thickness may influence the efficiency and accuracy of printed products. The choice of printing parameters often varies for different bioinks[46–48]. Therefore, we tried to determine the optimal parameters for printing the hydrogels in the optimal manner.

As shown in Fig. 3A, a flat cylinder used as the printing prototype was designed and tested with each layer separately exposed to blue light for 16, 24, 32, or 40 s. The prolongation in each layer's light exposure time led to increased total printing time and reduced efficiency (Fig. 3B). However, the appropriate extension of light exposure

time increased the printing accuracy of the cylinder by forming a tangent angle of adjacent planes closer to 90° (Fig. 3C). With the increase in G′, the mechanical strength of the created hydrogels was improved by the prolongation of light exposure time (Fig. 3D). Given the balance of printing rate, accuracy, and mechanical properties, a light exposure time of 32 s was chosen for this study.

Then, we printed the cylinder by adjusting the layer thickness from 0.1 to 0.4 mm (Fig. 3E). The increased layer thickness significantly reduced the printing time (Fig. 3F). Moreover, printing accuracy was slightly impaired due to the fact that the tangent angle of the adjacent planes was further away from 90° when the layer thickness was increased (Fig. 3G). There was a decline in the mechanical strength of the printed hydrogels when the layer thickness was increased (Fig. 3H). Taking into account the printing efficiency, accuracy, and mechanical properties, we selected a layer thickness of 0.2 mm for the following experiments.

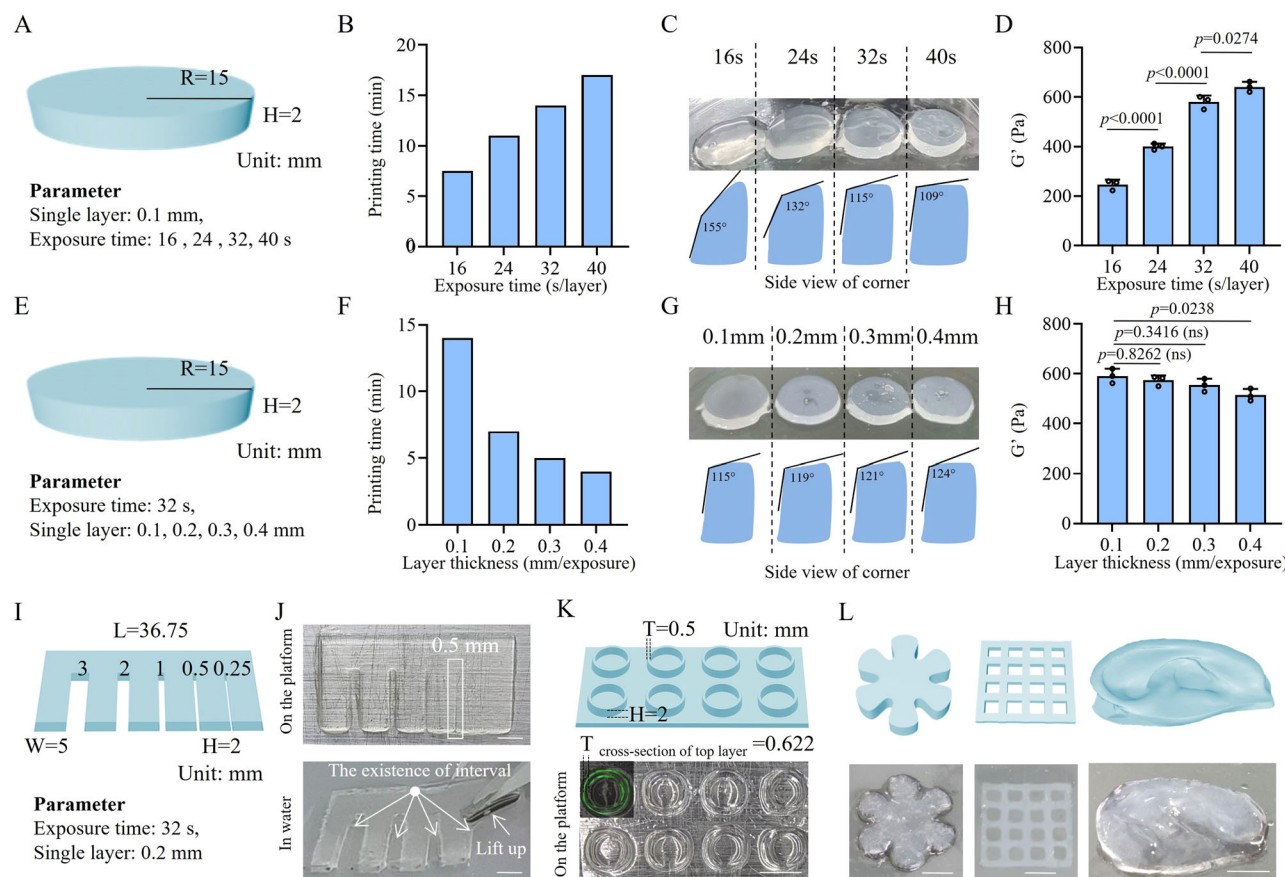

**Fig. 3 | DLP printability of the all-peptide hydrogel. A** Flat cylinder model for the determination of the optimal blue light exposure time per layer. **B** Prolongation of the light exposure time of each layer increased the printing time and reduced the efficiency. **C** Prolongation of the light exposure time of each layer increased the printing accuracy by forming a tangent angle of the corner closer to 90°. **D** Prolongation of the light exposure time of each layer increased the storage modulus (G') of the printed cylinder hydrogel. $n = 3$ independent experiments. **E** Flat cylinder model for the determination of the optimal printing layer thickness. **F** Increase in layer thickness significantly reduced the printing time. **G** Increase in layer thickness slightly impaired the printing accuracy by forming a tangent angle of the corner further away from 90°. **H** Increase in layer thickness slightly impaired the storage modulus (G') of the printed cylinder hydrogel. $n = 3$ independent

experiments. **I** Comb model for the determination of printing resolution. **J** Real printed comb with the printing resolution reaching 0.5 mm because the gap at 0.5 mm was the minimum distance required to separate adjacent comb teeth. Scale bars = 5 mm. **K** Printing microtube models with the wall thickness at 0.5 mm. The wall thickness of real microtubes was measured to be 0.622 mm, reflecting that the printing error was 0.122 mm. Scale bar = 5 mm. **L** DLP printing of customized objects in different shapes, including hexagonal petals, microporous scaffolds, and ear models. Scale bars = 5 mm. The $p$ values in the figures (**D** and **H**) are determined by one-way ANOVA followed by Tukey's multiple comparisons test. Data are presented as mean ± SD. ns, not significant. R radius, H height, W width, T thickness. Source data are provided as a Source Data file.

To further determine the resolution of DLP-printed hydrogels, we designed a comb model with the gradient distance of neighboring comb teeth ranging from 3 mm to 0.25 mm (Fig. 3I). The comb model was delicately printed on the platform with a small quantity of uncrosslinked bioink sticking to the surface. After removal of the redundant bioink by immersing the comb in water, the intervals of comb teeth were better visualized until the distance was shortened to 0.5 mm, at which the comb tooth was able to be lifted. When the interval distance was set as 0.25 mm, the gap between comb teeth disappeared. These results implied that the printing resolution could reach 0.5 mm (Fig. 3J). In order to figure out the printing error range, we designed a microtube array in which the wall thickness of the microtube was 0.5 mm. After printing, the wall of the microtube was measured as 0.622 mm under confocal microscopy by staining the wall using fluorescein isothiocyanate (FITC). Therefore, the printing error was calculated to be 0.122 mm (Fig. 3K). This finding could interpret the disappearance of the comb teeth's gap at the interval of 0.25 mm because the printing error range of adjacent comb teeth was accumulated to nearly 0.25 mm. Based on the resolution and fidelity of DLP printing of the γ-PGA hydrogel, differently-shaped objects were

printed, such as hexagonal petals, microporous scaffolds, and ear models (Fig. 3L).

## HUVECs*vegf165+* were co-printed using the all-peptide bioink and demonstrated satisfactory cell survival and proliferation

A specific lentivirus plasmid (pHS-AVC-1580) was constructed to infect HUVECs, which contained the transcripts encoding the restriction endonucleases of NheI and PvuII, the green fluorescence protein (GFP) of mNeongreen, puromycin N-acetyl-transferase for neutralizing puromycin, and VEGF 165 (Fig. 4A). The lentivirus plasmid (pHS-AVC-ZQ328) without the *VEGF 165* transcript was used as control. The gene sequence and structure of the VEGF 165-overexpressing lentivirus plasmid were verified by gene sequencing and DNA electrophoresis after digestion with NheI and PvuII. Figure 4B confirms the presence of *VEGF 165* transcript signals, and Fig. 4C shows that plasmid fragments conformed to their expected size after digestion.

After infection by the control and VEGF 165-overexpressing lentiviruses and selection with puromycin, HUVECs*vector* and HUVECs*vegf165+* were successfully generated with the percentage of GFP+ cells up to

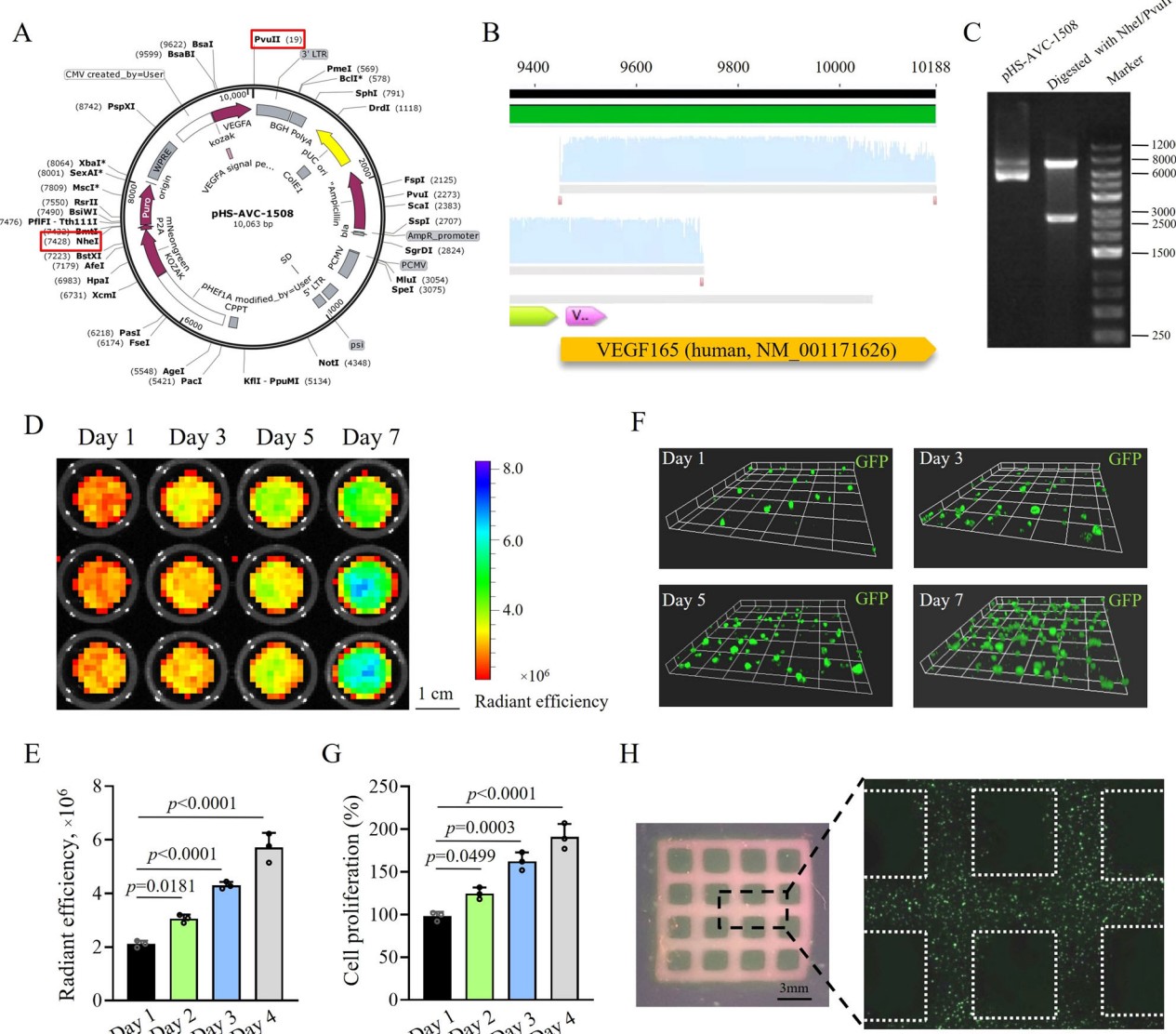

**Fig. 4 | Co-printability of HUVECs$^{vegf165+}$ using the all-peptide bioink. A** Structure of VEGF 165-overexpressing lentivirus plasmid, pHS-AVC-1580. Red frameworks mark the two restriction endonucleases of NheI and PvuII for cutting off the plasmid in the later validation experiment. **B** Gene sequencing identifies the VEGF 165 sequence in the plasmid. **C** Size of two fragments of VEGF 165-overexpressing lentivirus plasmid cut off by NheI and PvuII restriction endonucleases conforms to the expectations (e.g., 7409 bp and 2654 bp). Two independent experiments have confirmed similar results. **D** IVIS spectrum of HUVEC$^{vegf165+}$-laden hydrogel sheets when cultured for different time periods. **E** Quantitative analysis of the radiant efficiency of cell sheet IVIS images when cultured for different time periods. $n = 3$ independent experiments. **F** $Z$-axis stacking of the cell sheets using confocal microscopy when cultured for different time periods. **G** CCK-8 analysis of cell proliferation after different culture time periods. $n = 3$ independent experiments. **H** High-fidelity HUVEC$^{vegf165+}$-laden microporous scaffold embedded with homogeneous cells. The $p$ values in the figures (**E** and **G**) are determined by one-way ANOVA followed by Tukey's multiple comparisons test. Data are presented as mean ± SD. Source data are provided as a Source Data file.

>90% (Supplementary Fig. 6A, B). VEGF signals in HUVECs$^{vegf165+}$ were significantly increased compared to those in HUVECs$^{vector}$ (Supplementary Fig. 6C, D). Flat cylinder-shaped cell sheets were fabricated by co-printing HUVECs$^{vegf165+}$ ($1 \times 10^5$ cells/mL) with the all-peptide bioink. Following the culture of cell sheets, GFP intensity was gradually improved under the imaging of IVIS spectrum (Fig. 4D, E), implying that the cell transcriptional activity was maintained and cell counts may be increased. To further determine the cell proliferation, $z$-axis stacking on the cell sheets was performed using confocal microscopy, which revealed an increase in cell number (Fig. 4F). Moreover, the cell count kit-8 (CCK-8) assay also confirmed the apparent proliferation of HUVECs$^{vegf165+}$ in the cell sheets (Fig. 4G).

Additionally, we investigated the spatial controllability of cells based on the DLP-printed all-peptide bioink. As shown in Fig. 4H, a HUVEC$^{vegf165+}$-laden microporous hydrogel scaffold was printed, which presented a network structure. The confocal mosaic imaging of the scaffold further revealed that these HUVECs$^{vegf165+}$ were embedded in the network structure homogeneously.

## HUVEC$^{vegf165+}$-laden hydrogel released VEGF 165 in a self-renewable manner, promoted angiogenesis in vitro, and protected endothelial cells from HG-induced injuries via inhibiting Bax-elicited mitochondrial perforation

By loading living cells, the HUVEC$^{vegf165+}$-laden hydrogel was considered to be a living material with a remarkable self-renewal ability of VEGF 165, which is a splice variant of VEGFA that can be secreted extracellularly[49–51]. To verify this, we compared the releasing kinetics of VEGF 165 of HUVEC$^{vegf165+}$-laden hydrogel, HUVEC$^{vector}$-laden hydrogel,

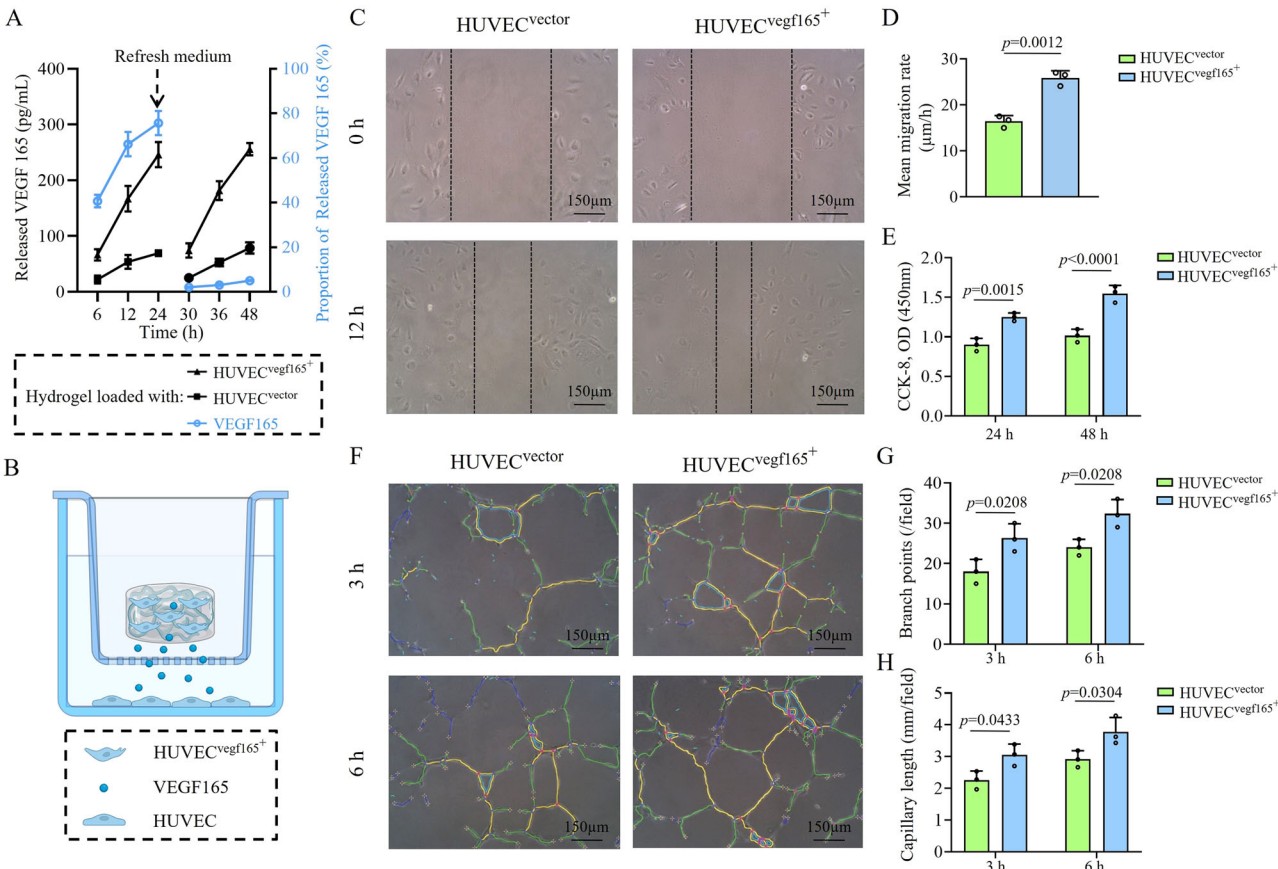

**Fig. 5 | Releasing kinetics of VEGF 165 by different hydrogel platforms and the resultant biological functions. A** HUVEC$^{vegf165+}$-laden hydrogel presented distinctive self-renewal ability and enhanced VEGF 165 release. The VEGF 165 releasing curves of cell-laden hydrogel in black were corresponding to the left $Y$-axis, while the curves of VEGF 165-loaded hydrogel in blue were in different releasing profiles and corresponding to the right $Y$-axis. $n = 3$ independent experiments. **B** Co-culture cell model for studying the biological effects of HUVEC$^{vegf165+}$-laden hydrogel. This figure was created with BioRender.com and has been granted a publication license. **C** Wound healing of HUVECs was accelerated following treatment with HUVEC$^{vegf165+}$-laden hydrogel. **D** Quantitative analysis of wound healing speed. $n = 3$ independent experiments. **E** HUVEC proliferation was improved by HUVEC$^{vegf165+}$-laden hydrogel. $n = 3$ independent experiments. **F** Tube formation of HUVECs was enhanced by HUVEC$^{vegf165+}$-laden hydrogel. **G** Quantitative analysis of branch points. $n = 3$ independent experiments. **H** Quantitative analysis of capillary length. $n = 3$ independent experiments. The $p$ values in the figure (**D**) and figures (**E**, **G**, and **H**) are determined by two-sided unpaired $t$ test and two-way ANOVA followed by Sidak's multiple comparisons test, respectively. Data are presented as mean ± SD. Source data are provided as a Source Data file.

and VEGF 165 recombinant protein-loaded hydrogel, during which the medium used to immerse the hydrogels was refreshed after the first 24 h. As shown in Fig. 5A, the cumulative released VEGF 165 in the three types of hydrogels was gradually increased with time, before refreshing the medium. The quantity of VEGF 165 released from the HUVEC$^{vegf165+}$-laden hydrogel was much greater than that released from the HUVEC$^{vector}$-laden hydrogel. Once the medium was refreshed, the release of VEGF 165 was significantly attenuated in the VEGF 165-loaded hydrogel, but the VEGF 165 releasing curves of cell-laden hydrogels were almost consistent with those before refreshing the medium. Altogether, these experiments reflected that the HUVEC$^{vegf165+}$-laden hydrogel could exert an enhanced paracrine action by releasing VEGF 165 in a self-renewable manner.

To further investigate the biological functions of self-renewable VEGF 165 released from the HUVEC$^{vegf165+}$-laden hydrogel, a co-culture cell model using the transwell chamber system was established, in which the cell-laden hydrogels were placed in the upper chamber and HUVECs were seeded in the bottom chamber (Fig. 5B). In comparison to the HUVEC$^{vector}$-laden hydrogel, the HUVEC$^{vegf165+}$-laden hydrogel promoted wound healing (Fig. 5C, D), cell proliferation (Fig. 5E), and tube formation of HUVECs in the bottom chamber (Fig. 5F–H), which suggested the potent angiogenetic competence of the HUVEC$^{vegf165+}$-laden hydrogel.

Since DM is always accompanied by vascular endothelial cell damage caused by HG[10,52], we further studied the effects of HUVEC$^{vegf165+}$-laden hydrogel on the HG-induced cell injuries. Figure 6A shows that HG, especially at a concentration of 40 mM, could lead to the death of HUVECs. Cell viability was not impaired in the osmotic control with 5 mM glucose and 35 mM D-mannitol. Then, we collected the cell supernatant of the HUVEC$^{vegf165+}$-laden hydrogel and HUVEC$^{vector}$-laden hydrogel and adjusted the glucose concentration to 40 mM to treat HUVECs (Fig. 6B). The results showed that the higher quantity of VEGF 165 released from the HUVEC$^{vegf165+}$-laden hydrogel alleviated HG-induced cell death (Fig. 6C) and reduced reactive oxygen species (ROS) production triggered by cell oxidative stress (Fig. 6D).

Previous studies have revealed that mitochondria are involved in ROS metabolism and mitochondrial dysfunction leads to ROS overproduction[53,54]. We found that the self-renewable VEGF 165 from the HUVEC$^{vegf165+}$-laden hydrogel could halt mitochondrial damage, thereby decreasing ROS production and improving cell viability. To verify this finding, we co-labeled mitochondria using MitoTracker and the DNA/RNA damage markers of 8-hydroxy-2'-deoxyguanosine (oxo8dG), 8-oxo-7,8-dihydroguanine (oxo8Gua), and 8-oxo-7,8-dihydroguanosine (oxo8G). The outcome of the immunofluorescence (IF) staining of the DNA/RNA damage markers indicated that they were almost co-localized with mitochondria when the cells were injured by

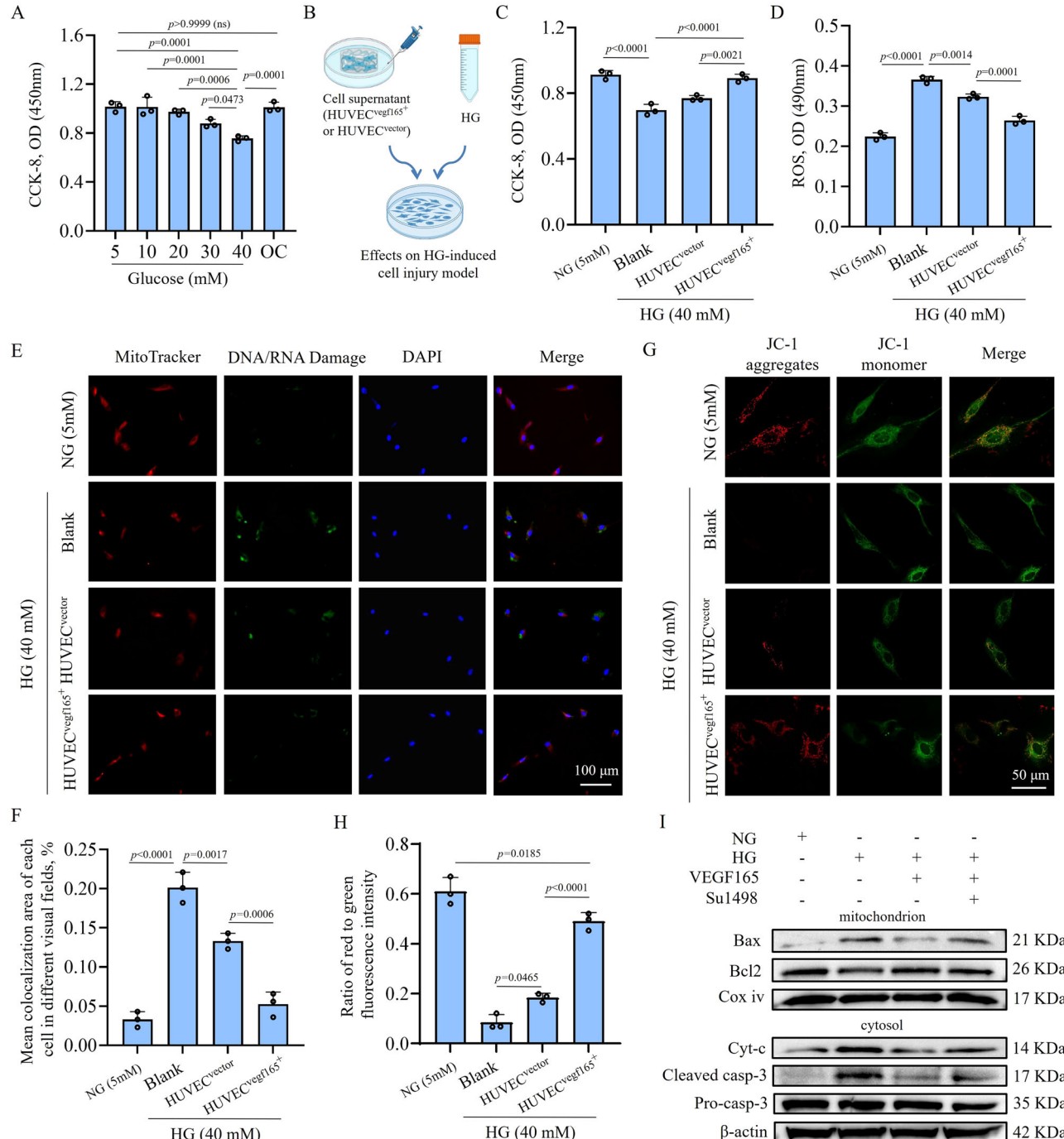

**Fig. 6 | Self-renewable VEGF 165 from HUVEC$^{vegf165+}$-laden hydrogel improves HG-induced endothelial cell injuries via protecting mitochondrial permeability.** **A** HG impairs the viability of HUVECs. OC, osmotic control with 5 mM glucose and 35 mM D-mannitol. $n = 3$ independent experiments. **B** Schematic diagram of cell models used to investigate the effects of the supernatant of cell-laden hydrogel platform on HG-induced HUVEC injuries. This figure was created with BioRender.com and has been granted a publication license. **C** Supernatant of HUVEC$^{vegf165+}$-laden hydrogel platform significantly improved the viability of HUVECs under the HG condition. $n = 3$ independent experiments. **D** Supernatant of HUVEC$^{vegf165+}$-laden hydrogel platform significantly reduced the ROS production of HUVECs under the HG condition. $n = 3$ independent experiments.
**E** Immunofluorescence (IF) staining assay revealed HG-induced DNA/RNA damages with the IF signals co-localized with mitochondria. The supernatant of HUVEC$^{vegf165+}$-laden hydrogel platform ameliorated the DNA/RNA damage. **F** Quantitative analysis

of mitochondrial DNA/RNA damage after different treatments. $n = 3$ independent experiments. **G** Supernatant of HUVEC$^{vegf165+}$-laden hydrogel platform improved the mitochondrial permeabilization detected with JC-1, a type of mitochondrial membrane potential (MMP) dye. **H** Quantitative analysis of the fluorescence intensity ratio of JC-1 aggregates to JC-1 monomers for determining MMPs after different treatments. $n = 3$ independent experiments. **I** Western blot results indicated that the VEGF 165 in the supernatant of HUVEC$^{vegf165+}$-laden hydrogel platform alleviated Bax-elicited mitochondrial perforation and casp-3-activated programmed cell death due to mitochondrial leakage. The results have been confirmed by two independent experiments. The $p$ values in the figures (**A**, **C**, **D**, **F**, and **H**) are determined by one-way ANOVA followed by Tukey's multiple comparisons test. Data are presented as mean ± SD. ns, not significant. Source data are provided as a Source Data file.

HG. When treated with the cell supernatant of HUVEC-laden hydrogels, the DNA/RNA damage signals were reduced, particularly for the HUVEC[vegf165+]-laden hydrogel, because this hydrogel could produce a greater quantity of VEGF 165 (Fig. 6E, F). The abovementioned evidence suggests that VEGF 165 from the HUVEC[vegf165+]-laden hydrogel alleviated HG-induced cell injuries via protecting mitochondria.

MMs play significant roles in maintaining mitochondrial homeostasis. The inner MMs are anchored with numerous vital enzymes, such as ATP synthase, to take part in redox and oxidative phosphorylation, which is accompanied by ROS production[55,56]. MM dysfunction has been reported to impair the enzymes on the MMs and cause leakage of mitochondrial contents to the cytosol, such as cytochrome c (Cyt-c), thus causing caspase activation and cell apoptosis[57]. Figure 6G, H shows that HG could damage MMs by reducing mitochondrial membrane potentials (MMPs) because the fluorescence intensity ratio of JC-1 aggregates to JC-1 monomers was significantly decreased. The treatment of the cell supernatant of HUVEC-laden hydrogels, particularly for the HUVEC[vegf165+]-laden hydrogel, elevated the MMPs and protected MMs.

To determine the molecular cues of changes in MMPs, we further detected Bax and Bcl2, the two members of B-cell lymphoma-2 family proteins. These two proteins are associated with the perforation of MMs and can be used to determine mitochondrial permeabilization based on the calculation of Bax/Bcl2 ratio in the separated mitochondria[58,59]. Western blot (WB) results shown in Fig. 6I and Supplementary Fig. 7 imply that HG could induce the recruitment of Bax to perforate MMs and release Cyt-c into the cytosol, thus activating caspase-3 (casp-3) by cleaving pro-casp-3, and causing cell apoptosis. Interestingly, the Bax-mediated disruption of MMs could be relieved by the treatment with the cell supernatant of the HUVEC[vegf165+]-laden hydrogel; in addition, programmed cell death was also improved. These protective functions mainly resulted from the released VEGF 165 because VEGFR-2 inhibition by SU1498[60] attenuated the beneficial effects of the cell supernatant. Thus far, we have clarified the advantages and mechanisms of the HUVEC[vegf165+]-laden hydrogel on the vascular endothelial cells.

## Diabetic wound closure of rats was augmented by HUVEC[vegf165+]-laden hydrogel

We evaluated the therapeutic effects of HUVEC[vegf165+]-laden hydrogel on the diabetic wounds of rats according to the experimental plan shown in Fig. 7A. The rat diabetic wound model was created by injecting streptozotocin (STZ) at a dose of 70 mg/kg intraperitoneally, followed by surgical removal of 1-cm-diameter full-thickness skin. The DM model was established successfully by presenting elevated blood glucose, retarded body weight growth, and increased water consumption during the entire experimental process (Supplementary Fig. 8), compared with normal control rats without injection of STZ. The three indictors of DM showed no significant differences among the groups treated with hydrogels, HUVEC[vector]-laden hydrogels, and HUVEC[vegf165+]-laden hydrogels. Day 3, day 7, and days 14 and 21 were the time points for tissue harvesting, which corresponded to the wound repair stages of inflammation, proliferation, and tissue remodeling[61]. Based on DLP printing, the hydrogels could be customized according to the irregular shapes of wounds (Supplementary Fig. 9A) and achieve high yield (Supplementary Fig. 9B).

The wound healing processes in the different groups are illustrated in Fig. 7B and re-depicted in Fig. 7C. The rats in the HUVEC[vegf165+]-laden hydrogel group exhibited a faster wound closure rate than those treated with hydrogels or HUVEC[vector]-laden hydrogels (Fig. 7D). In all groups, the hydrogels were degraded gradually until day 21 (Fig. 7E). VEGF 165 in the cell-laden hydrogels showed a regenerative manner, especially for the HUVEC[vegf165+]-laden hydrogel group in the first four days (Fig. 7F). Consistent with the in vitro findings, the production of VEGF 165 in vivo was much more in the HUVEC[vegf165+]-

laden hydrogel than that of HUVEC[vector]-laden hydrogel. This implied that HUVEC[vegf165+]-laden hydrogel could serve well as a good source of VEGF 165 for angiogenesis and wound healing.

Due to the enhanced performance of VEGF 165 released by the HUVEC[vegf165+]-laden hydrogels, the mitochondrial damages of vascular endothelial cells were relieved. As seen in Fig. 7G and Supplementary Fig. 10, single cell suspension (~600,000 cells in total) dissolved from wound tissues of three rats in different groups were prepared. 100,000 cells were isolated for cell staining of CD31-FITC and MitoSOX Red in turn, and then detected with flow cytometry. It was found that the proportion of vascular endothelial cells with mitochondrial damage in the HUVEC[vegf165+]-laden hydrogel group was 23.2%, which was much lower than the values of the simple hydrogel group (44.1%) and HUVEC[vector]-laden hydrogel group (41.8%). Moreover, a noninvasive measurement by laser speckle contrast analysis (LASCA) indicated that blood flow was increased due to improved tissue vascularization following treatment with HUVEC[vegf165+]-laden hydrogels compared with the other two groups (Fig. 7H, I).

Hematoxylin and eosin (HE) staining of the regenerated wound tissues in the three groups reflected the typical characteristics of rat skin healing, i.e., granulation tissue formation accompanied by epidermal contraction through crawling beneath the fibrous capsule (Fig. 7J)[62]. However, the thickness of granulation tissues, mainly consisting of immune cells, fibroblasts, microvessels, and collagen fibers, varied among the different groups. Specifically, the granulation tissues of diabetic wounds treated with HUVEC[vegf165+]-laden hydrogel were the thickest on days 7 and 14, compared to those of diabetic wounds that received other treatments, and almost reached the normal thickness on day 14 (Fig. 7K).

To eliminate the impacts of wound contraction on wound healing of rats, the edges of wounds were glued with silicone rings followed by interventions of simple hydrogel, HUVEC[vector]-laden hydrogel, and HUVEC[vegf165+]-laden hydrogel separately (Supplementary Fig. 11A). Generally, wound healing was postponed compared with that without silicone ring restrictions due to inhibition of wound contraction. Moreover, HUVEC[vegf165+]-laden hydrogel group demonstrated the fastest healing speed of diabetic wounds among the three groups (Supplementary Fig. 11B–D). Granulation tissues of diabetic wounds were covered by regenerated epidermal layers in all groups on day 10 (Supplementary Fig. 11E, F). The thickness of granulation tissues and epidermal layers (Supplementary Fig. 11G, H), and microvascular density (Supplementary Fig. 11I, J) were increased after treatment with HUVEC[vegf165+]-laden hydrogel compared with the other two groups. Altogether, it implied that the wound beds formed by the application of HUVEC[vegf165+]-laden hydrogels were more appropriate for re-epithelization and wound closure regardless of wound contraction.

## HUVEC[vegf165+]-laden hydrogel promoted the epithelialization, hair formation, scarless tendency, angiogenesis, and cell proliferation and metabolism of diabetic wounds

To comprehensively describe the wound-healing states, diverse tissue staining technologies were further applied. Re-epithelialization is important for wound healing. We used cytokeratin 14 (CK14), a marker of basal keratinocytes, and cytokeratin 10 (CK10), a marker of spinous keratinocytes, to evaluate the re-epithelialization on days 7 and 14. As shown in Fig. 8A, epithelial layers were encroached onto the wound bed and exhibited strong proliferative ability because the thickness far exceeded the epithelial layer thickness of the normal skin. In particular, on day 7, the thickness of the extended epithelial layer in the HUVEC[vegf165+]-laden hydrogel group was greater than that of the simple hydrogel group or HUVEC[vector]-laden hydrogel group, demonstrating the early activation of re-epithelialization by HUVEC[vegf165+]-laden hydrogel. In comparison, on day 14, the thickness of the extended epithelial layer in the HUVEC[vegf165+]-laden hydrogel group was the smallest and similar to that of normal skin, indicating rapid

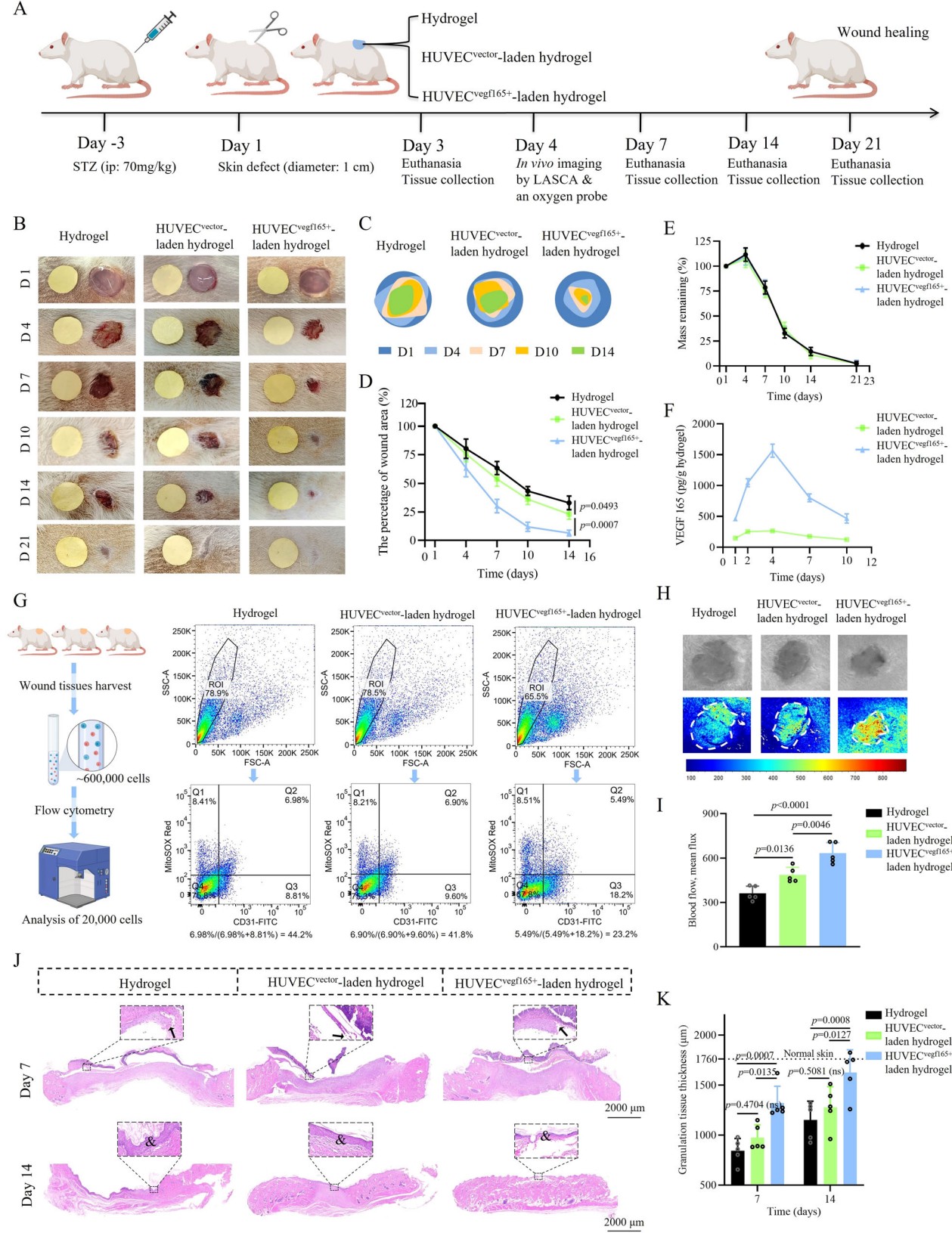

progression of epithelial proliferation to remodeling in this group. Moreover, on days 7 and 14, K14 staining of the hair follicles in the basal layer of the epidermis revealed a significant increase in the hair follicle density in the HUVEC$^{vegf165+}$-laden hydrogel group compared to hydrogel treatment and HUVEC$^{vector}$-laden hydrogel groups (Fig. 8B). These results demonstrate that the enhanced and self-renewable VEGF

165 release could effectively accelerate keratinocyte migration and hair follicle formation, consequently contributing to the re-epithelialization of the diabetic wounds.

To determine whether the improved wound healing was related to scar risks, we performed picrosirius red staining on wound collagen. Collagen type I stained in red cleaned the inflammatory sites, acted as a

**Fig. 7 | Evaluation of the effect of HUVEC$^{vegf165+}$-laden hydrogel on rat diabetic wounds. A** Experimental scheme. This figure was created with BioRender.com and has been granted a publication license. ip, intraperitoneal injection; LASCA, laser speckle contrast analysis. **B** Diabetic wound healing processes were recorded after the three treatments. Yellow paper as a size control is in a diameter of 1 cm. **C** Re-depiction of wound healing processes. **D** Comparison of wound closure rates following the three treatments. $n = 15$ rats/group on day 1; $n = 10$ rats/group on days 4 and 7; $n = 5$ rats/group on days 10 and 14. **E** Hydrogel degradation rate in the three treatment groups calculated by residue weight. $n = 5$ hydrogels/group. **F** Dynamic changes of VEGF 165 concentration in HUVEC$^{vector}$-laden hydrogels and HUVEC$^{vegf165+}$-laden hydrogels during the in vivo treatment process. $n = 5$ hydrogels/group. **G** Flow cytometry analysis of single cells lysed from wounds of three rats in each group indicated that the proportion of mitochondrial oxidative stress damage in vascular endothelial cells was 23.2% in the HUVEC$^{vegf165+}$-laden hydrogel group,

which was significantly lower than that of the hydrogel group (44.2%) and HUVEC$^{vector}$-laden hydrogel group (41.8%). MitoSOX red marked mitochondrial oxidative stress damage; CD 31 marked vascular endothelial cells. The left panel of the figure was created with BioRender.com and has been granted a publication license. **H** LASCA revealed increased blood flow following the treatment with HUVEC$^{vegf165+}$-laden hydrogels. **I** Quantitative analysis of blood flow based on mean flux. $n = 5$ rats/group. **J** HE analysis revealed varied degrees of granulation tissue formation and re-epithelization in the different treatment groups. Arrows: edges of the extended epithelial layer on day 7; &: formation of intact epithelial layer. **K** Quantitative analysis of granulation tissue thickness in the different treatment groups. $n = 5$ rats/group. The $p$ values in the figures (**D** and **K**) and figure (**I**) are determined by two-way ANOVA and one-way ANOVA, respectively, followed by Tukey's multiple comparisons test. Data are presented as mean ± SD. ns not significant. Source data are provided as a Source Data file.

chemo-attract for various cell types involved in wound healing, and increased the toughness, whereas collagen type III stained in green increased the elasticity; therefore, decrease of the collagen I/III ratio is beneficial for tissue flexibility and scar reduction[63–65]. As shown in Fig. 8C, the wounds in the HUVEC$^{vegf165+}$-laden hydrogel group presented a higher collagen I/III ratio on day 7 compared to hydrogel treatment and HUVEC$^{vector}$-laden hydrogel groups; however, the numerical value decreased rapidly to a level that was slightly lower than that of normal skin (-26) on day 14. On day 21 when wounds were almost healed, the collagen I/III ratio in the HUVEC$^{vegf165+}$-laden hydrogel group was still closest to the normal value (Supplementary Fig. 12). These findings suggest that the accelerated wound remodeling by HUVEC$^{vegf165+}$-laden hydrogel treatment did not induce scar formation.

On both days 7 and 14, compared to the hydrogel and HUVEC$^{vector}$-laden hydrogel groups, treatment with HUVEC$^{vegf165+}$-laden hydrogel increased the density of microvessels stained by CD 31 and α-SMA (Fig. 8D)[66], further verifying the potent angiogenetic effects of self-renewable VEGF 165 released from the HUVEC$^{vegf165+}$-laden hydrogel. The improved blood supply to the wound tissues further altered the cell proliferation and metabolism patterns of wounds. As shown in Fig. 8E, the regenerated granulation tissues in the HUVEC$^{vegf165+}$-laden hydrogel group presented denser proliferative cells distributed in the epidermis, dermis, and subcutaneous tissues on day 7. However, on day 14, the proliferative cells were mainly accumulated in the epidermis, and the density in the HUVEC$^{vegf165+}$-laden hydrogel group was lower than that in the other two groups, which was consistent with the observation of a thinner epithelial layer by CK14/CK10 staining in the HUVEC$^{vegf165+}$-laden hydrogel group on day 14. These findings further confirmed the beneficial effects on epithelial remodeling. Moreover, peroxisome proliferator–activated receptor gamma coactivator-1α (PGC-1α) is a biomarker of the cell metabolic rate[67], which had a higher expression in the HUVEC$^{vegf165+}$-laden hydrogel group on days 7 and 14 (Fig. 8F). This active state of cell metabolism was instrumental for promoting tissue proliferation and remodeling.

### Proteomics revealed that the improved outcomes of diabetic wound healing by HUVEC$^{vegf165+}$-laden hydrogel were associated with tissue vascularization and oxygenation, extracellular matrix (ECM) remodeling, and inflammatory modulation

To present an overview of the molecular signal changes that resulted from enhanced and self-renewable VEGF 165 stimulation, we further conducted proteomics detection for the diabetic wounds on day 7 after treatment with HUVEC$^{vegf165+}$-laden hydrogel. The HUVEC$^{vector}$-laden hydrogel group was used as control. In general, 500 proteins were significantly upregulated and 546 proteins were significantly downregulated between the HUVEC$^{vegf165+}$-laden hydrogel and HUVEC$^{vector}$-laden hydrogel (Fig. 9A). Figure 9B shows the protein expression differences between the two groups based on the principal component analysis (PCA), which suggested that the VEGF 165 over-expressed by HUVECs played a vital role in modulating wound repair

process. The volcano plot presented in Fig. 9C shows the differentially expressed proteins. The upregulated proteins labeled with the corresponding gene names were associated with the development, formation, and size enlargement of blood vessels (e.g., *Clic4*, *Eng*, *Enpep*, *Icam1*, *Itga1*, *Tgfbi*, *Cav1*, and *LOC108348074*), increase in ECM (e.g., *Efemp2*, *Col12a1*, *Lama4*, and *Lama5*), and enhanced cell adhesion (e.g., *Vcan*, *Sdc4*, *Itgb1*, and *Icam1*). The downregulated proteins were related to the reduction in markers of inflammation (e.g., *Alox5*, *Lbp*, *C5*, *Il36b*, and *Mmp8*) and bacterial infections (e.g., *Masp2*, *Mbl1*, *Krt34*, and *Cfd*).

We further performed functional enrichment analysis based on the differentially expressed proteins to uncover the dominant molecular pathways affected by the HUVEC$^{vegf165+}$-laden hydrogel. As seen in the biological process enrichment analysis on upregulated proteins (Fig. 9D), the HUVEC$^{vegf165+}$-laden hydrogel significantly promoted ECM assembly and organization, artery morphogenesis, and endothelial cell proliferation. In the biological process enrichment analysis on down-regulated proteins (Fig. 9E), the leukocyte aggregation and activation, neutrophil mediated immunity, and TNF production were significantly suppressed by the HUVEC$^{vegf165+}$-laden hydrogel. The specific proteins involved in the enrichment pathways mentioned above are listed in Supplementary Fig. 13. More interestingly, it was further revealed by a joint analysis of protein-protein interaction and protein-biological process relation that some differentially expressed proteins related to angiogenesis were simultaneously involved in inflammatory response and cell-matrix adhesion (Fig. 9F), which accounted for not only neo-vascularization, but also the other extensive biological functions such as inflammatory regulation and ECM remodeling that were achieved by the enhanced and self-renewable VEGF 165 from HUVEC$^{vegf165+}$-laden hydrogel. In addition, since we have revealed the effects of VEGF 165 on protecting mitochondrial injury in vitro, we further verified the in vivo function based on the proteome data. It was found that a total of 56 differentially expressed proteins were related to mitochondria (Supplementary Fig. 14). The heatmap shown in Fig. 9G marked these proteins where key proteins such as *Sod2* and *Hmgsc2* that reduced mitochondrial oxidative stress were upregulated in HUVEC$^{vegf165+}$-laden hydrogel group, and some proteins like *Mtco2* and *Cox5b* associated with mitochondrial damages were downregulated[68,69]. Protein-protein interaction and protein-biological process relation network analysis further indicated that oxidation-reduction process of mitochondria was a therapeutic target of VEGF 165 by HUVEC$^{vegf165+}$-laden hydrogel (Fig. 9H). Altogether, the proteomic results indicated extensive molecular signal changes in terms of inflammation, ECM remodeling, and mitochondrial oxidative stress in the diabetic wounds induced by the angiogenetic effect of HUVEC$^{vegf165+}$-laden hydrogel.

It has been reported that early resolution of inflammation is beneficial for scarless repair and wound healing[70,71]. To further confirm the finding that the HUVEC$^{vegf165+}$-laden hydrogel reduced inflammation, we assessed proinflammatory and chemotactic cytokines in diabetic wounds on day 7. As shown by a proteome profiler rat cytokine

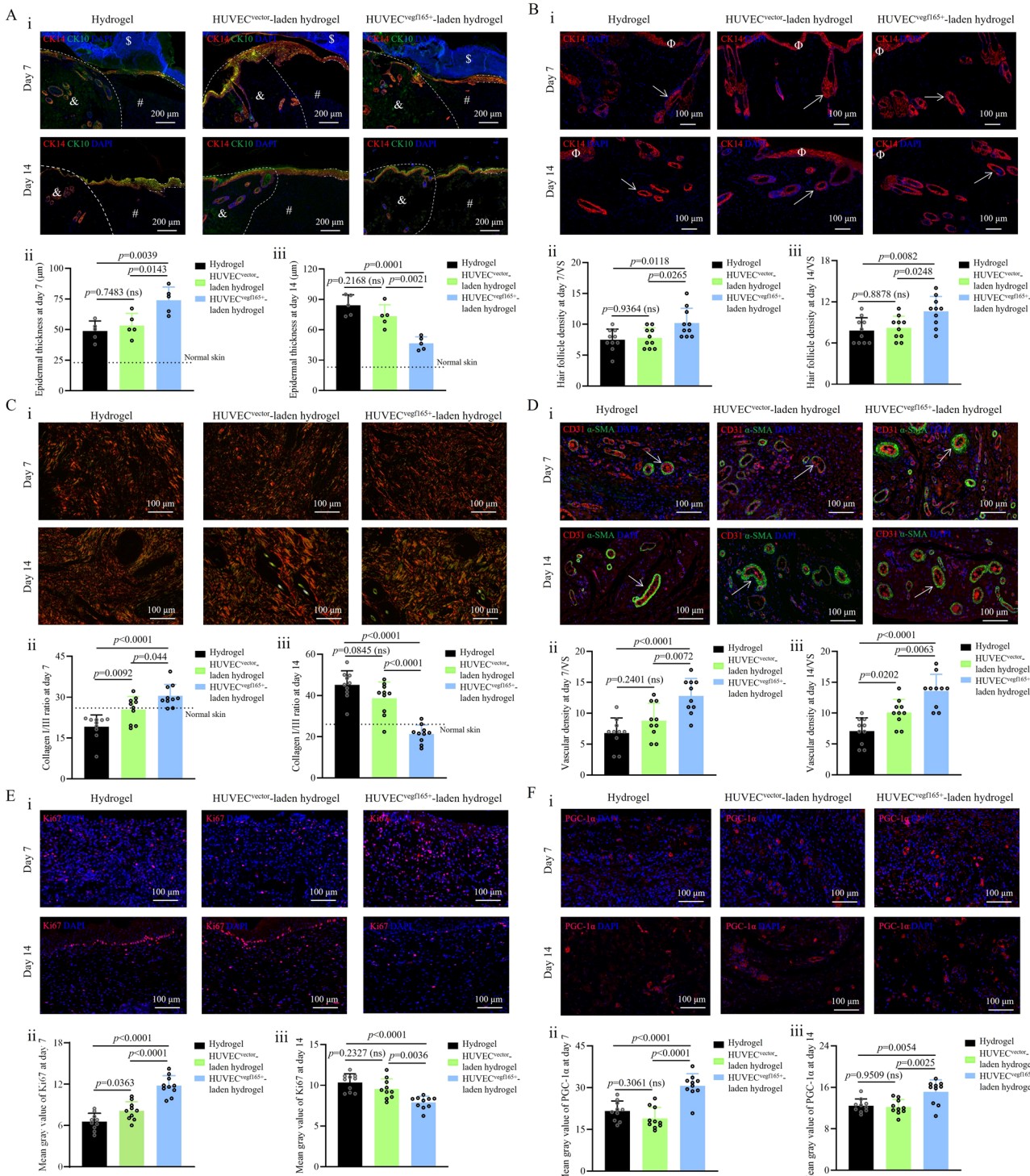

**Fig. 8 | Histological analysis of the healing process of diabetic wounds with different treatments. A** CK14/CK10 immunofluorescence (IF) staining of the epithelial layer. $: fibrous capsule; #: regenerated granulation tissue; &: edges of normal skin; dotted line: boundary separating normal skin and wound; dot-and-dash line: extended epithelial layer that covers the wound bed. $n = 5$ rats/group. **B** CK14 IF staining of hair follicles. Φ: epithelial layer; arrows: hair follicles. VS, visual field. $n = 10$. **C** Picrosirius red staining of collagen in the wounds. $n = 10$. **D** Vascular structure stained with CD 31 and α-SMA. Arrows: microvessels. $n = 10$. **E** Staining of proliferative cells by Ki67. $n = 10$. **F** Evaluation of cell metabolic rate by PGC-1α staining. $n = 10$. For **A**–**F**, i: representative tissue staining images; ii: quantitative analysis of corresponding wound images on day 7; iii: quantitative analysis of corresponding wound images on day 14. For **B**–**F**, $n = 10$ because two random visual fields were selected for each tissue sample harvested from five rats in each group. The $p$ values in each figure (ii and iii) are determined by one-way ANOVA followed by Tukey's multiple comparisons test. Data are presented as mean ± SD. ns, not significant. Source data are provided as a Source Data file.

array in Fig. 10A, B, the HUVEC$^{vegf165+}$-laden hydrogel presented decreased expression of various proinflammatory and chemotactic cytokines, such as CINC-1, CINC-2α/β, CINC-3, IL-1α, and LIX, compared to the hydrogel and HUVEC$^{vector}$-laden hydrogel groups. Moreover, in the HUVEC$^{vegf165+}$-laden hydrogel group, the expression of metalloproteinase inhibitor-1 (TIMP-1), which inactivates metalloproteinase-9, hinders tissue remodeling, and promotes hypertrophic scar formation, was lower than that in the other two groups. It supports the

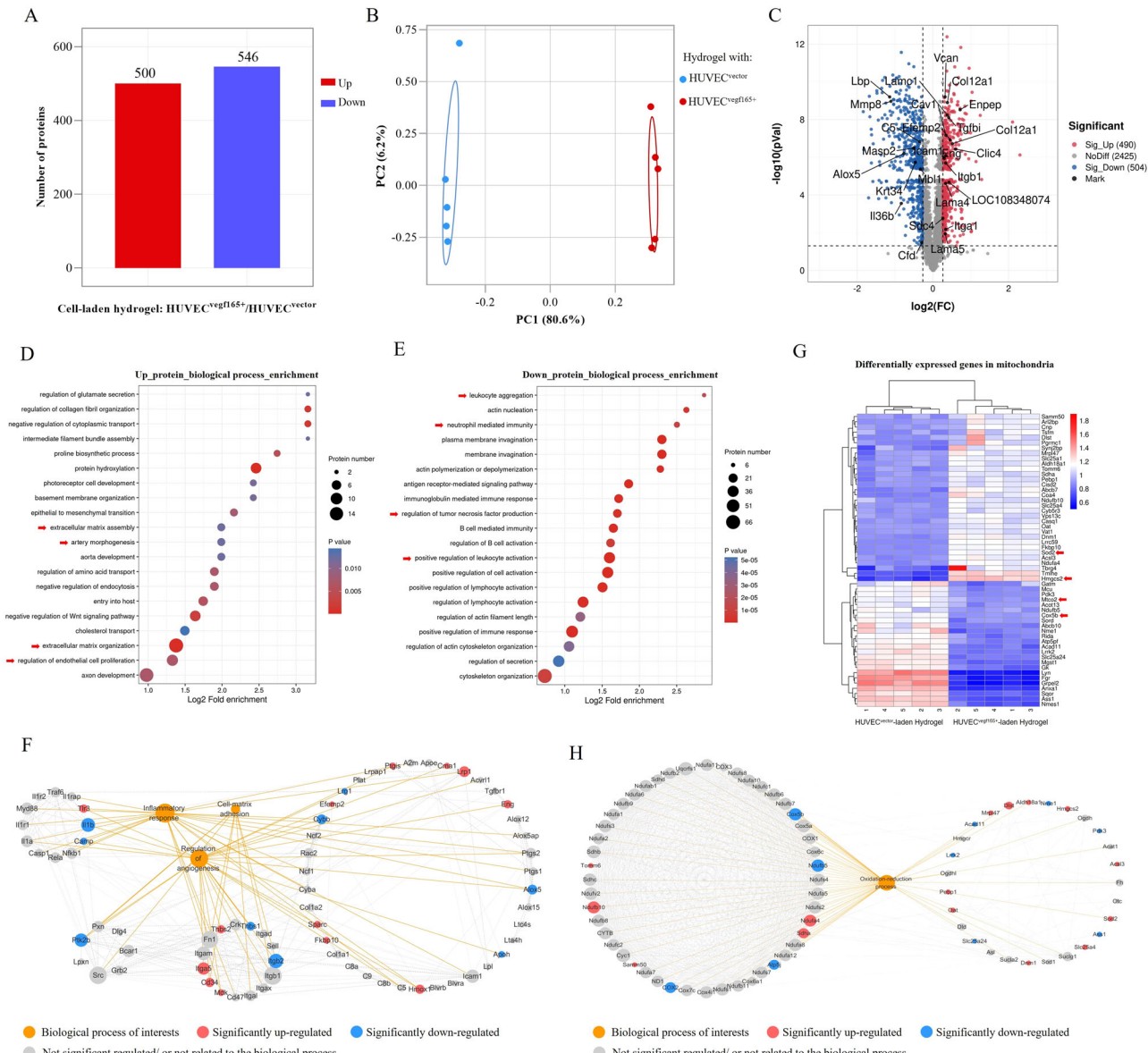

**Fig. 9 | Proteomics comparison of diabetic wounds on day 7 between HUVEC$^{vegf165+}$-laden hydrogel and HUVEC$^{vector}$-laden hydrogel. A** Number of differentially expressed proteins. **B** PCA reveals evident separation based on differentially expressed proteins between the two groups. $n = 5$ rats/group. **C** Volcano plot shows the differentially expressed proteins of interests. **D** Upregulated pathways in the biological process enrichment analysis. Arrows: upregulated pathways of interests regarding extracellular matrix deposition and vessel regeneration. **E** Downregulated pathways in the biological process enrichment analysis. Arrows: downregulated pathways of interests regarding inflammatory response. **F** Protein-protein interaction and protein-biological process relation network analysis

indicates that different biological processes during wound healing including inflammatory response, angiogenesis, and cell-matrix adhesion are interrelated based on some common differentially expressed proteins. **G** Heatmap shows the differentially expressed mitochondrial proteins of interests. Arrows: representative proteins related to mitochondrial oxidative stress damage. **H** Protein-protein interaction and protein-biological process relation network analysis reveals the key differentially expressed mitochondrial proteins involved in the oxidation-reduction process. The $p$ values in the figure (**C**) and figures (**D** and **E**) are determined by two-sided unpaired $t$ test and Fisher's exact test, respectively.

histological analysis results that the HUVEC$^{vegf165+}$-laden hydrogel promotes tissue remodeling and reduces scar risk. The VEGF level of diabetic wounds was elevated due to the production from the HUVEC$^{vegf165+}$-laden hydrogel.

Vascular formation provides oxygen transmission pipelines for oxygenation, which is a key factor determining tissue hypoxia, inflammation, and wound-healing rates[72]. Therefore, we measured the oxygenation of the diabetic wounds on day 4 using a luminescent oxygen probe, tris(4,7-diphenyl-1,10-phenanthroline) ruthenium (II) dichloride complex[73]. As shown in Fig. 10C, the oxygen signals were stronger in the HUVEC$^{vegf165+}$-laden hydrogel group, which suggested improved wound oxygenation due to the sustained VEGF 165 release

from the hydrogel. Accordingly, hypoxia inducible factor-1α (HIF-1α), an indicator of oxygen tension[74], exhibited lower transcriptional levels in the HUVEC$^{vegf165+}$-laden hydrogel group throughout the wound-healing process (Fig. 10D). HIF-1α has been reported to drive the expression of proinflammatory cytokines in various immune cells[75]. For this reason, we further explored malondialdehyde (MDA) and TNF-α levels to evaluate the oxidative stress and inflammatory state in diabetic wounds, respectively. It was found that MDA and TNF-α levels were significantly decreased in the HUVEC$^{vegf165+}$-laden hydrogel group (Fig. 10E, F). Thus far, it has been clarified that self-renewable and sustained stimulation by HUVEC$^{vegf165+}$-laden hydrogel prevents mitochondrial damages of vascular endothelial cells from

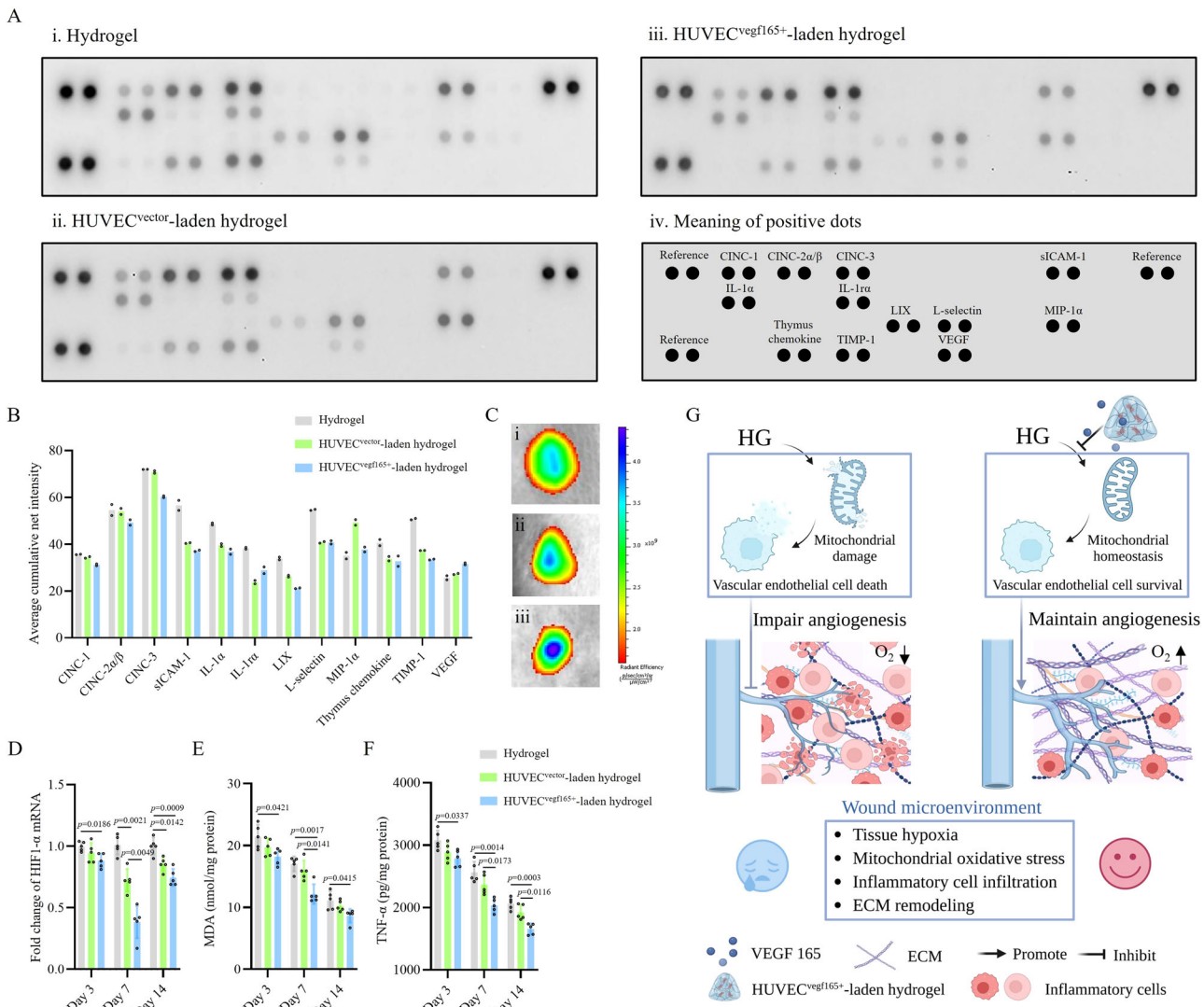

**Fig. 10 | HUVEC$^{vegf165+}$-laden hydrogel ameliorates oxygenation and inflammation of diabetic wounds. A** Proteome profiler rat cytokine array of diabetic wounds on day 7 with different treatments. **B** Quantitative analysis of the signal intensity of cytokines. $n = 2$ dots for each cytokine in the membrane. **C** IVIS spectrum of luminescent oxygen probe for the detection of oxygen contents of diabetic wounds on day 4. i: hydrogel; ii: HUVEC$^{vector}$-laden hydrogel; iii: HUVEC$^{vegf165+}$-laden hydrogel. **D** Comparison of HIF-1α transcriptional level among the diabetic wounds with different treatments on days 3, 7, and 14. $n = 5$ rats/group. **E** Comparison of MDA concentration among the diabetic wounds with different treatments on days 3, 7, and 14. $n = 5$ rats/group. **F** Comparison of TNF-α production among the diabetic

wounds with different treatments on days 3, 7, and 14. $n = 5$ rats/group.
**G** Therapeutic mechanism of wound healing by HUVEC$^{vegf165+}$-laden hydrogel. VEGF 165 was continuously released from the HUVEC$^{vegf165+}$-laden hydrogel to rescue HG-induced vascular endothelial cell death by inhibiting mitochondrial oxidative stress, thus improving tissue angiogenesis and oxygenation, and creating an eligible microenvironment for diabetic wound healing. This schematic diagram was created with BioRender.com and has been granted a publication license. The $p$ values in the figures (**D**–**F**) are determined by two-way ANOVA followed by Tukey's multiple comparisons test. Data are presented as mean ± SD. Source data are provided as a Source Data file.

HG and facilitates angiogenesis in diabetic wounds, which creates a satisfactory microenvironment by improving tissue hypoxia, reducing inflammation, and promoting ECM remodeling, thus ultimately contributing to diabetic wound healing (Fig. 10G).

## Discussion

In this study, we developed a tissue-friendly, high-efficiency, and accurate 3D printed all-peptide hydrogel platform based on the thiol-ene click reaction using γ-PGA-GMA, γ-PGA-SH, and RGDC. This platform could be constructed in a few minutes, reached a resolution power of 0.5 mm and a printing error of 0.122 mm, and offered stably-bound RGD for cell adhesion, which fulfill the requirements of clinically applicable biological scaffolds. Using angiogenesis as a target therapeutic point, we further modified this platform as a living material by loading functional HUVECs that have enhanced and self-renewable

VEGF 165 release, thereby solving the problems of growth factor loss or inactivation during wound applications.

Furthermore, our experiments on HUVECs revealed that VEGF 165 was a potent stimulator of angiogenesis in terms of cell migration and tube formation. Under the HG condition, VEGF 165 inhibited Bax-elicited mitochondrial perforation through interaction with VEGFR-2, which impeded the activation of casp-3 and prevented cell death. The discovery of this molecular mechanism underlines the distinctive functions of VEGF 165 for diabetic wound healing.

By building rat diabetic wound models, we demonstrated the therapeutic effects of HUVEC$^{vegf165+}$-laden hydrogel on accelerating wound closure, vascular formation, granulation tissue remodeling and regeneration, and re-epithelization. The scarring tendency of wounds treated with HUVEC$^{vegf165+}$-laden hydrogel was relatively low. Proteomics sequencing was used to describe the general changes of

molecular signals caused by the HUVEC$^{vegf165+}$-laden hydrogel, which reduced wound inflammation and improved angiogenesis, ECM remodeling, and cell adhesion. Rapid regression of inflammation is a positive factor contributing to the progression from the inflammatory phase to the proliferative phase and subsequent remodeling. Improved oxygenation of diabetic wound supported by well-formed microvessels may be responsible for the reduction of inflammation when the HUVEC$^{vegf165+}$-laden hydrogel was applied, which suppressed the HIF-1α transcriptional activity and reduced the production of downstream proinflammatory factors.

Based on our results, the HUVEC$^{vegf165+}$-laden hydrogel is an effective living material to deal with diabetic wounds. However, limitations exist in the printing resolution and rat models. The potential approaches that may improve the printing resolution include the increase in light source resolution or the addition of specific biomacromolecules to inhibit free radical polymerization at unexposed areas during photoinitiation[76,77]. Moreover, diabetic wound models of rats created by injection of STZ only represent part of complex diabetic wound pathophysiology in human bodies; therefore, additional trials on large mammals or patients are needed to perform to validate the treatment and determine a specific usage.

## Methods

### Materials
γ-PGA (Mw = 700 kDa, 95%) was obtained from Shineking Biotechnology Co., Ltd (Nanjing, China). 1-Ethy-3-(3-dimethylaminopropyl)-carbodiimide (EDC, 98%), N-hydroxy-succinimide (NHS, 98%), L-Cysteine hydrochloride (Cys·HCl, 98%), glycidyl methacrylate (GMA, 97%), 2-(N-morpholino)ethanesulfonic acid (MES, 99%), LAP (purity ≥ 98.0%), and hydrochloric acid (HCl, 37%) were obtained from Aladdin Industrial Corporation (Shanghai, China). RGDC with a purity of 98.46% was custom-made by NJPeptide (Nanjing, China). Dialysis membranes (MWCO 12 kDa) were obtained from Yuanye Bio-Technology Corporation (Shanghai, China). The lentiviral particles used for VEGF 165 overexpression or as a control vector were provided by SyngenTech (Beijing, China).

### Synthesis and printability of γ-PGA-SH, γ-PGA-GMA, and all-peptide hydrogel
**Synthesis of γ-PGA-SH.** γ-PGA (1.00 g, 7.75 mmol) was dissolved in 80 mL of MES buffer (0.1 M MES, 0.1 M NaCl, pH 5.0). EDC (2.89 g, 11.60 mmol) and NHS (1.73 g, 11.60 mmol) were added to the MES buffer and reacted for 30 min to activate the carboxylic group of γ-PGA at room temperature. Then, Cys·HCl (1.72 g, 11.6 mmol) dissolved in 20 mL of new MES buffer was added dropwise to the mixture. The pH of the reaction mixture was adjusted to 4.75 by the addition of 0.1 M HCl solution, and then, the reaction mixture was purged with nitrogen and kept stirring in a dark environment at room temperature (25−30 °C) for 24 h. Afterward, the reaction mixture was exhaustively dialyzed against acidic deionized water (pH 3.5) for 72 h. After dialysis, γ-PGA-SH was lyophilized to obtain a white powder (0.87 g, 87% yield).

**Synthesis of γ-PGA-GMA.** γ-PGA (1.00 g, 7.75 mmol) was dissolved in DI water to obtain a 1 wt% solution at room temperature. Subsequently, GMA was added to the solution at a 3:1 molar ratio of GMA to -COOH (of γ-PGA) gently under stirring. The pH value of the reaction solution was adjusted to 4.7 by adding 0.1 M HCl solution, followed by reaction at 60 °C for 8 h. Then, the reaction solution was transferred to a dialysis tube and purified using exhaustive dialysis against deionized water for 3 days. Finally, the dry product (0.93 g, 93% yield) was obtained by freeze-drying.

**Fabrication of the all-peptide hydrogel.** γ-PGA-SH, γ-PGA-GMA, RGDC, and LAP were dissolved in DI water based on the formula presented in Supplementary Table 1. The solution was crosslinked by exposure to blue light ($n = 405$ nm) for 20 s. The resultant hydrogels were termed as Gels 1, 2, 3, and 4 based on the increased polymer concentrations. The modified polymers were identified with $^1$H NMR and FTIR spectra. The obtained hydrogels were fully characterized with the swelling ratio, degradation, morphological scanning, and rheological and mechanical tests. Detailed procedures of each characterization are described in Supplementary Note 1.

**Printability of the all-peptide hydrogel.** The premixed solution based on the formula of Gel 3 was chosen for printing using a DLP printer (ANYCUBIC Photon Ultra) because Gel 3 presented relatively high compressive strength and fracture energy. The objects to be printed were designed using C4D software (Maxon Computer, Germany). The layer thickness and blue light ($n = 405$ nm) exposure time were regulated to optimize the printing parameters evaluated on the printing efficiency and resolution. The printed products were separated from the platform using a sharp blade. The tangent angle of the adjacent planes in the printed cylinders was calculated based on the side-view images by using Solidwork software (Dassault Systèmes Corp, France). The printing error range was determined by comparing the microtube model's real size measured using z-axis stacking of a confocal microscope (LSM 980; ZEISS, Germany) with the designed size. FITC (HY-66019; MCE) solution was used to stain the wall of microtubes.

### Generation of HUVEC$^{vector}$-laden hydrogel and HUVEC$^{vegf165+}$-laden hydrogel
**Generation of HUVECs$^{vegf165+}$ and HUVECs$^{vector}$.** The lentiviral particles used for VEGF 165 overexpression or control were provided by SyngenTech, which were synthesized by following the manufacturing process reported previously[78]. The structure of VEGF 165-overexpressing lentivirus was composed of the lentiviral transfer plasmid (pHS-AVC-1508), described as pLV-hef1a-mNeongreen-P2A-Puro-WPRE-CMV-VEGF165 (human, NM_001171626), and the lentiviral packaging vectors (SyngenTech, Beijing, China). The control lentiviral transfer plasmid (pHS-AVC-ZQ328) did not contain the VEGF 165 sequence. The lentiviral transfer plasmids were examined by gene sequencing and DNA electrophoresis after digestion with NheI and PvuII (SyngenTech, Beijing, China). HEK 293FT cells (SyngenTech, Beijing, China) were used to package the lentiviral particles. The VEGF 165-overexpressing lentivirus and control vector lentivirus were titrated to concentrations of $2.89 \times 10^8$ TU/mL and $2.75 \times 10^8$ TU/mL, respectively. Before transfection, $1 \times 10^6$ HUVECs (P2-P4; Sciencell, USA) cultured in the ECM medium (Sciencell, USA) were seeded to a cell culture bottle (430720; Corning, USA). On the next day, the prepared lentivirus was added to the culture medium at the multiplicity of infection (MOI) of 10. Polybrene (SyngenTech) at 8 µg/mL was used to increase transfection efficiency. After 18 h, the culture medium was refreshed every 3 days. Three days later, puromycin (P8230; Solarbio, Beijing, China) at 4 µg/mL was added for the selection of transfected cells. On day 7, the resultant HUVECs$^{vegf165+}$ and HUVECs$^{vector}$ were imaged under a fluorescence microscope (Invitrogen, USA), and the proportion of fluorescent cells was calculated using ImageJ (NIH, USA). The VEGF expression of HUVECs$^{vegf165+}$ and HUVECs$^{vector}$ was identified using the VEGF antibody (1:50, ab155944; Abcam) and imaged with fluorescence microscopy. The mean gray value was determined by ImageJ software (NIH, USA).

**Construction of cell-laden hydrogel platform.** The bioink was prepared by dissolving the substances in the formula of Gel 3 and phosphate-buffered saline (PBS) with 10 w/v% fetal bovine serum (FBS; Gibco, USA). HUVECs$^{vegf165+}$ and HUVECs$^{vector}$ were digested with trypsin cell digestion solution (KeyGEN Biotech, Nanjing, China) and then added to the bioink at a concentration of $1 \times 10^5$ /mL. The cell-laden hydrogels were printed by adding the bioink to the DLP printer. Layer thickness and light exposure time per layer were set as 0.2 mm and

32 s, respectively. The produced HUVEC$^{vegf165+}$-laden hydrogels were then cultured in the complete ECM medium for 7 days. During this period, the hydrogels were scanned using an IVIS imaging system (XRMS Series III, PerkinElmer, USA) at a wavelength of 488 nm. Moreover, based on the confocal microscopy, Z-axis stacking and mosaic imaging of the HUVEC$^{vegf165+}$-laden hydrogels were performed to visualize the embedded cells. Cell proliferation on different days was determined by the ratio of cell count on that day to the cell count on the first day in three random visual fields.

### Study of VEGF 165 release kinetics and biological functions

**Description of VEGF 165 release curves.** VEGF 165-loaded hydrogels were prepared by dissolving VEGF 165 (Pepro Tech, USA) in the precursor solution of Gel 3 at a concentration of 1 µg/mL and were then printed in the shape of a cylinder with a diameter of 10 mm and a height of 3 mm. The HUVEC$^{vector}$-laden hydrogel and HUVEC$^{vegf165+}$-laden hydrogel were prepared in the same shape. All hydrogels ($n = 3$) were immersed in 10 mL of incomplete DMEM (KeyGEN Biotech, Nanjing, China). At each pre-set time point (6, 12, 24, 30, 36, and 48 h), 100 µL of leach liquor was collected for detection of VEGF 165 concentration using an enzyme-linked immunosorbent assay (ELISA) kit (Enzyme-linked Biotechnology Co., Ltd, Shanghai, China). At 24 h, the culture medium was refreshed. The release curves of the cell-laden hydrogels were drawn based on the VEGF 165 concentration, and the release kinetics of VEGF 165-loaded hydrogels were determined by the ratio of the released VEGF 165 to the total quantity of the protein added to the hydrogels.

**Study of the angiogenetic functions.** The transwell cell model was established to perform the wound healing, CCK-8, and tube formation assays. In particular, for wound healing assay, $8 \times 10^3$ HUVECs were added to the lower chamber of a 24-well transwell plate (Corning, USA) and cultured in the complete DMEM with 2.5 w/v% FBS. Scratches were created using a 20-µm pipette, and the detached cells were removed by washing with PBS. Cell-laden hydrogels were printed in the shape of a cylinder with a diameter of 5 mm and a height of 3 mm and were placed into the upper chambers and cultured in 200 µL of complete DMEM with 2.5 w/v% FBS. After 12 h, cell migration was observed with a microscope (CKX53, Olympus, Japan), and the mean migration rate was calculated by dividing the cell migration distance by the time. As for the CCK-8 assay, $8 \times 10^3$ HUVECs were added to the lower chamber of a 24-well transwell plate and cultured in the complete DMEM with 10 w/v% FBS. Cell-laden hydrogels were placed in the upper chambers and cultured in 200 µL of complete DMEM with 10 w/v% FBS. At 24 and 48 h, the HUVECs in the lower chamber were stained with the CCK-8 reagent (KeyGEN Biotech, Nanjing, China) and the optical density (OD) was measured at a wavelength of 450 nm. Moreover, the tube formation assay was performed according to our protocol reported previously with slight modifications[79]. Briefly, $5 \times 10^4$ HUVECs were added to the Matrigel (356231; BD, USA)-coated lower chamber of a 24-well transwell plate and cultured in complete DMEM with 10 w/v% FBS. Simultaneously, cell-laden hydrogels were placed in the upper chambers and cultured in 200 µL of the complete DMEM. After 3 or 6 h, tube formation was observed under a microscope and analyzed using ImageJ software based on three different visual fields.

**Study of protective roles on HG-induced mitochondrial damage and cell death.** The complete DMEM with different concentrations of glucose (5, 10, 20, 30, and 40 mM) was used to treat HUVECs to investigate HG-induced cell death. To determine the influence of osmotic pressure on cells, an osmotic control was set, which contained 5 mM glucose and 35 mM D-mannitol (M1902; Sigma, USA). Two days later, cells were incubated with CCK-8 reagent for 1 h, and the OD value was measured at 450 nm. Moreover, the leach liquor was prepared by immersing cell-laden hydrogels into incomplete DMEM at a volume ratio of 1:4 for 2 days to study the cellular and molecular effects. As shown in Fig. 6B, the leach liquor was mixed with HG-containing DMEM to produce the working medium, in which the glucose concentration was 40 mM, FBS concentration was 10 w/v%, and VEGF 165 concentrations of HUVEC$^{vector}$-laden hydrogel and HUVEC$^{vegf165+}$-laden hydrogel were almost 1.04 ng/mL and 5.3 ng/mL, respectively. The different complete DMEM with normal glucose (NG) of 5 mM or HG with different leach liquor (i.e., blank, HUVEC$^{vector}$-laden hydrogel, and HUVEC$^{vegf165+}$-laden hydrogel) was used to treat HUVECs for 2 days, followed by CCK-8 assay and ROS probe (KeyGEN Biotech, Nanjing, China) detection at 488 nm. Moreover, the HUVECs after the different treatments were co-stained with MitoTracker Deep Red (Warbio, Nanjing, China) and anti-DNA/RNA damage antibody (1:200, ab183393; Abcam), and then observed under a fluorescence microscope. The MMP detection was performed using a commercial JC-1 kit (KeyGEN Biotech, Nanjing, China) followed by cell imaging with confocal microscopy. To further reveal the molecular mechanisms and verify the contributing factor of VEGF 165 for the protective functions, different complete DMEM with NG (5 mM) or HG with different leach liquors (i.e., blank and HUVEC$^{vegf165+}$-laden hydrogel), or HG with the leach liquor of HUVEC$^{vegf165+}$-laden hydrogel and SU1498 (5 µM, MCE, USA), a specific inhibitor of VEGFR-2, was separately used to treat HUVECs for 2 days. The separation of mitochondria and cytosol was achieved by using a commercial kit (3601; Beyotime Biotechnology, China), following the manufacturer's instructions. The molecules of our interests were detected by WB using the antibodies against Bax (1:2000, ab32503; Abcam), Bcl2 (1:2000, ab182858; Abcam), Cox iv (1:2000, ab202554; Abcam), Cyt-c (1:5000, ab133504; Abcam), cleaved casp-3 (1:500, ab32042; Abcam), Pro-casp-3 (1:1000, ab32150; Abcam), and β-actin (1:1000, ab8226; Abcam). The detailed WB procedures are described in Supplementary Note 2.

### Therapeutic effects of HUVEC$^{vegf165+}$-laden hydrogel on the diabetic wound model of rats

**General analysis of wound closure rate.** Eighty-two seven- to eight-week-old male Sprague Dawley rats weighing almost 220 g were provided by Huachuang Sino Co., Ltd. The rats were housed in individual cages with free access to sterile water and chow, controlled temperature, and natural light–dark cycles. Before creating diabetic wounds, 79 rats were intraperitoneally (ip) injected with STZ (Yuanye Biotech, Shanghai, China) at 70 mg/kg, and 3 rats were used as a normal control without injection of STZ. Three days later, the 79 rats were randomly divided into hydrogel group ($n = 23$), HUVEC$^{vector}$-laden hydrogel group ($n = 28$), or HUVEC$^{vegf165+}$-laden hydrogel group ($n = 28$). Each group was created with 10-mm full-thickness skin defects, followed by treatment with DLP printing of hydrogels, HUVEC$^{vector}$-laden hydrogels, or HUVEC$^{vegf165+}$-laden hydrogels in the shape of a cylinder with a diameter of 10 mm and a height of 3 mm. The hydrogels were fixed with a transparent Tegaderm dressing film (3 M, USA), and not replaced during the entire experimental process. The images of diabetic wounds were captured using a smart phone (Vivo, China) on days 1, 4, 7, 10, 14 and 21 by referring to a round paper with a diameter of 10 mm. The percentage of wound area was calculated by the ratio of the area on that day to the area on day 1. Meanwhile, the remained hydrogels on wounds were weighed. To verify the VEGF 165 self-generating ability of cell-laden hydrogel, a small part of hydrogel in five rats of HUVEC$^{vector}$-laden hydrogel group and HUVEC$^{vegf165+}$-laden hydrogel group was harvested and then weighed on days 1, 2, 4, 7, and 10. VEGF 165 concentration in these hydrogel fragments was measured with the ELISA kit. Moreover, on days 3, 7, 14, and 21, a portion of rats in each group were euthanized, and the regenerated wound tissues were collected for detailed analysis. The weight, water consumption, and blood glucose level (Yuwell Medical Equipment, China) of partial rats were measured every day. After completing the entire experiment, one of the three rats in the normal control was used to verified wound shape

adaption by creating wounds in the shape of triangle, rectangle, and heart. Hydrogels in corresponding shapes were printed and applied on wounds to test the customizability.

In addition, wound healing of rats is always accompanied by wound contraction which is much more prominent in rats than that in human beings; therefore, we suppressed wound contraction of rats by fixation of silicone rings to wound edges using a 502 glue (Deli, China)[80]. Except this, the methods of creating diabetic wounds and interventions were the same for a total of 30 seven- to eight-week-old male Sprague Dawley rats with 10 rats in each group. The wounds were imaged on days 1, 7, 10, 14 and 21. Moreover, on day 10, five rats of each group were euthanized, and the regenerated wound tissues were collected for subsequent analysis. The animal protocols above were approved by the Animal Investigation Ethics Committee of Jinling Hospital (2020JLHSKJDWLS-123).

**In situ detection of tissue vascularization and oxygenation.** To investigate the change and relation of blood supply and tissue oxygenation following different treatments, five rats of each group were randomly chosen on day 4 for the evaluation of neovascularization based on the LASCA using moorFLPI-2 (Moor Instruments, UK). On the same day, oxygenation of rat diabetic wounds was detected using a luminescent oxygen probe, tris(4,7-diphenyl-1,10-phenanthroline) ruthenium (II) dichloride complex (Alfa Aesar, UK), with an IVIS spectrum imaging system (XRMS Series III).

**Measurement of mitochondrial damage in vascular endothelial cells dissociated from wound tissues.** On day 7, the regenerated granulation tissues of three rats in each group were harvested and stored in the sCelLive™ Tissue Preservation Solution (Singleron, China) on ice. The specimens were washed with Hanks Balanced Salt Solution (HBSS) for three times, minced into small pieces, and then digested with 3 mL sCelLiVE™ Tissue Dissociation Solution by Singleron PythoN™ Tissue Dissociation System at 37 °C for 15 min. The cell suspension was collected and filtered through a 40-micron sterile strainer. Afterwards, the GEXSCOPE® red blood cell lysis buffer (Singleron) was added, and the mixture [Cell: RCLB = 1:2 (volume ratio)] was incubated at room temperature for 5 min to remove red blood cells. The mixture was then centrifuged at $300 \times g$ for 5 mins to remove supernatant and suspended softly with PBS. It was found by cell counting that around 600,000 cells were finally obtained for each group, in which 100,000 cells were isolated for cell staining with the vascular endothelial cell marker, CD31-FITC (1:100, ab33858; Abcam) and the mitochondrial damage marker, MitoSOX Red (HY-D1055; MCE) in turn according to the manufacturers' instruction. BD LSRFortessa™ X-20 flow cytometer was used to analyze 20,000 cells.

**Histological analysis.** Half of the collected wound tissues were immersed in 4% polyformaldehyde for 24 h, followed by dehydration and wax leaching. The wax-soaked tissues were embedded in the embedding machine (Wuhan Junjie Electronics Co., Ltd, China) and sliced into 4-μm-thick sections. HE staining was performed to measure the thickness of regenerated diabetic wound tissues. Picrosirius red staining was used to quantify the collagen I/III ratio. IF staining was performed to mark the epithelial layer and hair follicle with CK10 (1:1000, GB111413; ServiceBio) and CK14 (1:1000, MA5-11599, Invitrogen; 1:1000, GB11803, ServiceBio), to label the vascular structure using CD31 (1:100, MA5-16951; Invitrogen) and α-SMA (1:500, GB111364; ServiceBio), to reflect cell proliferation based on Ki67 (1:500, GB111141; ServiceBio), and to compare the cell metabolic rate by the staining of PGC-1α (1:100, ab106814; Abcam). The sections were then incubated with the corresponding secondary antibodies, and the nuclei were stained with DAPI. Quantification of the IF staining signals was performed on two random visual fields of each sample of the three groups using ImageJ software.

**Proteomics analysis of regenerated wound tissues.** Approximately one-third of the wound tissues of each sample were used for proteomics analysis by Lc-Bio Technologies, Hangzhou, China. Briefly, the proteins were extracted by tissue homogenization using a tissue lyser (60 Hz, 2 min), followed by 15-min centrifugation (20,000 × $g$, 4 °C). Quality control of protein extraction was conducted using the Bradford method and SDS-PAGE. Then, the obtained proteins were digested with trypsin at the 50:1 ratio of protein to trypsin. After digestion, the peptides were labeled with Tandem Mass Tag (TMT; ThermoFisher Scientific, USA). Equal quantities of peptides from each sample were mixed and fractionated using a 3.5 μm 4.6 × 150 mm Agilent ZORBAX 300Extend-C18 column on the UltiMate™ 3000 Binary Rapid Separation System (Thermo Fisher Scientific, USA). The gradient elution was performed using monitoring elution peaks at 214 nm, and fractions were collected every minute. The obtained fractions were freeze-dried. Afterward, the dried peptide samples were redissolved and separated using the EASY-nLC™ 1200 system (Thermo Fisher Scientific, USA). The separated peptides were ionized using a nano-ESI and then transferred to Orbitrap Exploris™ 480 mass spectrometer (MS; Thermo Fisher Scientific, USA) for DDA mode detection. MaxQuant (version 2.1.4.0) software was used to analyze the TMT-plexed MS/MS raw data. Statistical analysis was performed using R software (version 4.0.0). Proteins were considered statistically different when the $p$ value was <0.05 and the fold-change was >1.2. OmicStudio tools at https://www.omicstudio.cn/tool were used to draw principal component analysis chart, volcano plot, biological process enrichment scatter plot, and heatmap[81]. STRING database at https://cn.string-db.org/ was used to study protein-protein interactions where the combined score å 0.700. Cytoscape at https://cytoscape.org/ was applied to draw protein-protein interaction and protein-biological process relation networks.

**Cytokine and metabolite detection.** Multiple cytokines of the wound tissue homogenates were measured simultaneously by using a Proteome Profiler Rat Cytokine Array Kit, Panel A (ARY008, R&D, USA) strictly according to the manufacturer's instructions. The concentrations of the lipid peroxide products (MDA), and the proinflammatory cytokines (TNF-α) were detected using an MDA kit (KGT003, KeyGen Biotech) and TNF-α ELISA kit (Enzyme-linked Biotechnology Co., Ltd, Shanghai, China).

**qPCR analysis.** Quantitative PCR was performed according to the laboratory protocol[82]. In general, the total RNA was extracted from the wound tissues and reverse-transcribed with HiScript III RT SuperMix (Vazyme Biotechnology, China). Then, the reverse transcription products were amplified by QuantStudio™ 3 (Applied Biosystems, USA) using SYBR Green QPCR Master Mix (Vazyme Biotechnology, China) by referring to the manufacturer's instructions. The primer sequences of the genes (*HIF-1α* and *Tubulin*) synthesized by General Bio Co., Ltd (Chuzhou, China) are presented in Supplementary Table 2[83]. *Tubulin* was used as a housekeeping gene, and mRNA expression was calculated by the $2^{-\Delta\Delta Ct}$ method and normalized to the average value of the simple hydrogel group.

### Statistical analysis
Unless stated otherwise, all statistical analyses were performed with GraphPad Prism 9.0 (GraphPad Software, USA), and represented as mean ± standard deviation (SD). Numerical data from the repetitive measurements were analyzed by using the unpaired Student's $t$ test (two-sided), one-way ANOVA (≥3 groups), or two-way ANOVA (with two influencing factors). Multiple comparisons were performed if necessary. A $p$-value of less than 0.05 indicates statistical significance.

### Reporting summary
Further information on research design is available in the Nature Portfolio Reporting Summary linked to this article.

## Data availability

The proteomics data are publicly accessible at iProX (https://www.iprox.cn/page/home.html) with the dataset identifier PXD036045. Source data are provided with this paper. Data are available from the authors upon request.

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

## Acknowledgements

We are grateful for financial support from the National Natural Science Foundation of China [82300648 (J.H.), 82270595 (J.R.), and 32171402 (Z.L.)], China Postdoctoral Science Foundation [BX20220393 (J.H.), and 2022M723891 (J.H.)], Natural Science Foundation of Jiangsu Province [BK20231091 (J.H.)], Key Research and Development Program of Jiangsu Province [BE2022823 (J.R.), and BE2023810 (B.C.)], and Jiangsu Provincial Medical Innovation Center [CXZX202217 (J.R.)].

## Author contributions

J.H., X.W., B.C., and J.R.: conceptualization; J.H., Z.L., B.C., and J.R.: funding acquisition; J.H., R.Y., P.W.: material synthesis and characterization; J.H. and Z.L.: DLP printing; J.H., J.J., Z.L., Y.L., S.L., C.C., G.Q., and K.C.: cell and animal experiments; X.W., B.C., and J.R.: supervision and validation; J.H.: writing the original draft; J.H., R.Y., J.J., and X.W.: review, editing, and revision.

## Competing interests

The authors declare no competing interests.
