## [Peer Review File · Nature Communications]

A Click Chemistry-mediated All-peptide Cell Printing Hydrogel Platform for Diabetic Wound HealingREVIEWER COMMENTS

Reviewer #1 (Remarks to the Author):

This manuscript presents a 3D printable hydrogel matrix for cell printing consistent with cell-based therapies. It does so using cross-linking chemistry that has been used before, yet with novel materials, and that is a valuable addition to the biomaterials literature. It does so in the context of cell-based delivery of VEGF-A for promotion of angiogenesis in the context of diabetic wound healing, and that is a valuable addition to the wound healing literature. My comments on the paper have more to do with its presentation and impact than with the details of its approach (which one important exception, having to do with the animal model).

1. From the perspective of the materials work, the main contribution of the study, the work is well done, but not well presented in the context of the literature. Materials that can be cured in situ have been presented based on photopolymerization (very early work by JA Hubbell), based on addition reactions for cross linking after mixing two components (works by JA Hubbell, MP Lutolf, JA Burdick, D Seliktar, KS Anseth), and by photo-induced cross linking (used in this work, developed by KS Anseth). The previous work by Anseth is cited somewhat, but this present work should be set more in the context of the other work as well. The novelty here is only in the use of the gamma-polyglutamic acid material.
2. From the perspective of the biological approach, there are many examples of cellular and protein engineering approaches to deliver VEGF-A as well as VEGF-C in diabetic wound healing, indeed which have been very successful and are much easier to translate than the approach of a cellular delivery to express the VEGF-A. This work is essentially overlooked. Examples come from A Banfi, JA Hubbell, SA Eming, MM Martino, and several others. Thus, the impact of this work is rather modest, since it is known that sustained presence of VEGF-A either alone or in combination with other stabilizing growth factors such as PDGF-BB (see work by A Banfi, who even found optimal ratios of VEGF-A and PDGF-BB) is beneficial. To take a more complicated approach is not of high impact.
3. The animal model for wound healing is problematic. Use of STZ to induce diabetes is fine, although the animals are then relatively freshly diabetic. One more commonly uses the db/db mouse for these studies, but still in principle the STZ rat is fine. The main problem is that wound healing in these animals involves a substantial contraction contribution, not just re-epithelialization. One usually then splints the wounds, in which a relatively rigid ring is glued to the skin surrounding the wound, to prevent contraction. This does not seem to have been done here. The results on angiogenesis induction are very clear from the results, but it is not possible to separate the healing effects of contraction from re-epithelialization.

Reviewer #2 (Remarks to the Author):

I have reviewed the manuscript entitled "Self-renewable VEGF 165 Generated by Click Chemistry-mediated All-peptide Cell Printing Hydrogel Platform for Diabetic Wound Healing via Improving Mitochondrial Function" by Jinjian Huang et al., which describes a novel approach for diabetic wound healing using a self-generating hydrogel platform. Overall, I find this work to be of high quality and the results to be of significant interest for the field of regenerative medicine. However, there are several points that need to be addressed before the manuscript can be considered for publication.

1. The authors established a high glucose-induced injury model and determined the appropriate glucose concentration using the CCK-8 assay. However, they did not consider the effect of osmotic pressure changes on cell viability. I suggest the authors conduct additional experiments using mannitol to exclude the effect of osmotic pressure on cell viability.
2. In Figure 5, the time points for Wound healing and Tube formation assays are not consistent. The authors should clarify why different time points were chosen for these experiments.
3. In Figure 6i, the internal reference bands are inconsistent, and the cleaved caspase-3 band is not clear. The expression levels of Cyt-c, Bax, and Bcl2 do not match with the provided internal control β -action. Please provide new data with appropriate internal controls. I recommend the authors to provide more reliable and comprehensive data.
4. The authors selected a glucose concentration of 40 mM for the high glucose model, and it is unclear whether the osmotic pressure at this concentration affects cell proliferation. I suggest the authors investigate the effect of osmotic pressure on cell proliferation under 40 mM glucose conditions.
5. In Figure 7, the authors compare the wound healing rate at different time points and evaluate blood vessel formation only on day 4 using LASCA analysis. However, in line 604-605, the authors state that tissue samples were collected on days 3, 7, and 14. It is unclear why they did not do LASCA analysis on days 7 and 14. The authors should provide more detailed information about their experimental design.
6. The authors explain that collagen remodeling can reflect scar formation from lines 340-349. However, in Figure 7B, the wound is not completely healed on day 14. I suggest the authors provide additional data on the complete wound healing time (such as day 17 or day 21) to more fully explain their findings.
7. Although Figure S8 provides graphs for blood glucose, body weight, and water consumption, it only shows the results from day 1 and does not provide a dynamic change over the entire 14-day period. The authors should provide data for the entire 14-day period.

Overall, this manuscript presents a novel approach for diabetic wound healing using a self-generating hydrogel platform. However, the authors need to address the above points to improve the manuscript's clarity, accuracy, and overall scientific rigor. I recommend major revisions to this manuscript.

Xuqiang Nie, Ph.D, Prof.

Reviewer #3 (Remarks to the Author):

Huang L. et al., in their manuscript “Self-renewable VEGF 165 Generated by Click Chemistry-mediated All-peptide Cell Printing Hydrogel Platform for Diabetic Wound Healing via Improving Mitochondrial Function” reported on the preparation of blue light-induced 3D-printable hydrogel loaded with VEGF 165-transfected HUVEC for wound healing. This study demonstrates satisfactory experimental quality and complete identification. However, there exist many issues throughout the whole manuscript that need to be resolved/explained, especially the inappropriate relevance between the materials design and the applications. Experiments and data presented in the work were loosely connected. This reviewer does not believe that the current version of the manuscript provides sufficient novelty and quality for publication in Nature Communications. Here are the major comments.

1. The composition of the hydrogel used in this study is highly similar to the previous publication by the authors (International Journal of Biological Macromolecules 142 (2020): 332-344), merely with the addition of extra RGDC to the hydrogel system in this study. However, it was demonstrated in previous work that the γ -PGA-GMA/ γ -PGA-SH hydrogel is sufficient to embed bone marrow-derived MSCs with good viability. With this case, why is it necessary to add the extra RGDC in the present study? Furthermore, if adding the RGDC in the γ -PGA-based hydrogel is insufficient, would the all-peptide hydrogel be necessary? Besides, the authors have previously published HA/ γ -PGA-based hydrogels, while the differences between the hydrogels in this study and the previous hydrogels need to be clarified.
2. The modulus of living skin ranges from 10-100 kPa (Nature Reviews Materials 5.5 (2020): 351-370), while the hydrogel studied in this article has a modulus of 0.1-1 kPa, which may be considered too soft. Would it be better to use a material with a modulus closer to that of skin?
3. The authors indicated that Gel 3 was chosen for evaluating DLP printability due to its improved mechanical properties. What is the basis for selecting from Gels 1, 2, and 3? Are there higher concentration groups, such as Gels 4, 5, and 6?
4. A resolution power of 1 mm appears insufficient to guarantee high-precision printing. Could the Pr value, U value, or other printing parameters (On the progress of hydrogel-based 3D printing: Correlating rheological properties with printing behavior. International Journal of Pharmaceuticals, 121506,2022) of all-peptide hydrogels be quantified?
5. What is the self-renewal ability of the HUVECvegf165+-laden hydrogel and how was it compared with other hydrogels? Meanwhile, what is the release mechanism by which the self-renewable VEGF 165 released from the HUVECvegf165+-laden hydrogel promotes wound healing, cell proliferation, and tube formation of HUVECs in vitro? The authors should add more discussion about this point, which the authors highlighted.
6. What is the mechanism of HUVECvegf165+-laden hydrogel on re-epithelialization, hair formation, scarless tendency, angiogenesis, and cell proliferation and metabolism of diabetic wounds? The authors only showed significant results but did not give enough discussion between self-renewable VEGF165 and these effects.
7. The description and the target application (wound dressing or artificial skin) of the hydrogel platform in wound healing experiments in vivo is not clear. Has the HUVECvegf165+-laden hydrogel platform ever been replaced within 14 days? If not, the degradation of the hydrogel is important. The authors should

show the status of remaining hydrogels after attaching chronic wounds within 14 days. Also, the viability of VEGF 165-transfected HUVEC in the 3D-printed hydrogel should be specifically demonstrated. If the HUVECveg165+ is unable to survive up to 14 days in the hydrogel covering the wounds, the self-renewal ability of VEGF 165 highlighted in this work should be explained in detail.

8. The wound exudate of chronic wounds is a serious problem. Would the exudate influence the release efficiency or the function of VEGF 165 from the hydrogel? Could this hydrogel system absorb excessive wound exudate? If not, it may cause infection when the wound exudate accumulated.

9. The authors mentioned that the wounds were followed by treatment with DLP printing of hydrogels. However, the round wound model is not complicated to highlight the necessity of using DLP printing. In Figure 7B, the hydrogels on the wound as shown appear to be bulk hydrogel rather than 3D-printed hydrogel. The reliability and quality of histological figures are not enough. The data seemed doubtful and may mislead the readers.

10. This reviewer strongly recommends that the authors re-clarify the linkages between each part of this study. The current version of the manuscript fails to clearly state convincing reasons for the selection of materials, the necessity of 3D-printed hydrogel in animal studies and the discrepancy in effectiveness, and the doubts about the use of HUVECveg165+-laden hydrogel in wound dressings. The mechanism by which the HUVECveg165+-laden hydrogel may affect mitochondrial function is a key feature in this study, yet it does not seem to be highlighted in the article and is insufficient to bring this study up to the standard of Nature Communications.

Reviewer #4 (Remarks to the Author):

In this study, Huang and colleagues employ a method of fabricating a 3D printed all-peptide hydrogel platform that aids the wound healing process and promotes angiogenesis in the context of diabetes. With the complexity of the disease etiology and the increased rate of diabetic individuals in developing foot ulcers and prolonged infection; diabetes reduces the ability of the skin to repair itself thus dysregulating the wound healing process. The manuscript clearly states this rational. That said, while the results are noteworthy, the below comments and considerations would enhance the quality and impact of this article.

1. The use of biomaterials such as hydrogels as drug carriers for skin regeneration and repair advanced was applied 20 years ago, so the originality on the use of hydrogels is moderate. However, the concept of fabricating a biocompatible and biodegradable delivery system that can easily adapt to the wound shape and release the exact amount of the required growth factors to support the wound healing process is novel as it hasn't been considered in the literature. In addition to looking at the mechanistic work of this paper, the authors have presented the reduction in the inflammatory state of the wound and enhanced mitochondria function, which support their finding and hypothesis.

2. To make the paper more appealing to a broader audience, the definitions on click reaction (line 116) and digital light processing technique should be mentioned in the introduction.

3. In section 2.2, the DLP variable parameters light exposure and layer thickness were considered, are there other parameters that would affect the scaffold properties such as printing speed, polymers flowrate and ambient parameters?

4. To strengthen the paper concept, it is recommended to expand in the discussion section to include recent publications on this topic, in addition to explaining the clinical effectiveness of the hydrogel platform for wound healing. Below are some suggestions:

<https://link.springer.com/article/10.1007/s12274-022-4192-y>

<https://www.sciencedirect.com/science/article/abs/pii/S1742706122003233>

<https://www.sciencedirect.com/science/article/abs/pii/S1001841722007100>

<https://www.frontiersin.org/articles/10.3389/fbioe.2019.00342/full>

In conclusion, the manuscript fits within the journal theme and is recommended from publication once the above points are considered.

Responses to Reviewers:

To Reviewer #1:

General impression: This manuscript presents a 3D printable hydrogel matrix for cell printing consistent with cell-based therapies. It does so using cross-linking chemistry that has been used before, yet with novel materials, and that is a valuable addition to the biomaterials literature. It does so in the context of cell-based delivery of VEGF-A for promotion of angiogenesis in the context of diabetic wound healing, and that is a valuable addition to the wound healing literature. My comments on the paper have more to do with its presentation and impact than with the details of its approach (which one important exception, having to do with the animal model).

Our response: We really appreciate your time and efforts on reviewing our manuscript, and thank you very much for your recognition on this research. A point-to-point response to your comments are attached below for your evaluation.

Comment 1. From the perspective of the materials work, the main contribution of the study, the work is well done, but not well presented in the context of the literature. Materials that can be cured in situ have been presented based on photopolymerization (very early work by JA Hubbell), based on addition reactions for cross linking after mixing two components (works by JA Hubbell, MP Lutolf, JA Burdick, D Seliktar, KS Anseth), and by photo-induced cross linking (used in this work, developed by KS Anseth). The previous work by Anseth is cited somewhat, but this present work should be set more in the context of the other work as well. The novelty here is only in the use of the gamma-polyglutamic acid material.

Response 1. This is a helpful suggestion that reminds us to describe a more comprehensive research background. Therefore, we have revised the “Introduction part”. Some classic literatures by JA Hubbell and the colleagues were added such as Reference 22, which bound growth factors to synthetic matrix to gain bioactivities for wound repair. In addition, we supplemented one more meaning of γ -PGA-based hydrogel from the perspective of all-peptide hydrogel (lines 113-120, page 6). Compared with traditional all-peptide hydrogel crosslinked of methacrylate gelatin, methacrylate silk, or methacrylate decellularized extracellular matrix which present uncontrollable variability in their composition between batches due to the differences in sources and processing methods, the γ -PGA-based hydrogel with a definite chemical composition can not only serve well as cell scaffolds, but also ensure the results to be more reproducible and stable during applications. Moreover, brief introductions on cell vehicles for drug delivery and DLP printing have been added in lines 104-106, page 6, and lines 110-113, page 6, respectively. With these changes on Introduction part, this study can be understood by readers from a more objective and comprehensive perspective.

Comment 2. From the perspective of the biological approach, there are many examples of cellular and protein engineering approaches to deliver VEGF-A as well as VEGF-C in diabetic wound healing, indeed which have been very successful and are much easier

to translate than the approach of a cellular delivery to express the VEGF-A. This work is essentially overlooked. Examples come from A Banfi, JA Hubbell, SA Eming, MM Martino, and several others. Thus, the impact of this work is rather modest, since it is known that sustained presence of VEGF-A either alone or in combination with other stabilizing growth factors such as PDGF-BB (see work by A Banfi, who even found optimal ratios of VEGF-A and PDGF-BB) is beneficial. To take a more complicated approach is not of high impact.

Response 2. Thank you very much for this critical comment! We completely agree that increasing evidence has verified the effectiveness of growth factors such as VEGF-A and PDGF-BB. However, a meta-analysis on clinical trials of growth factors revealed that the efficacy of growth factors for diabetic wound healing remains questionable and varies with their category, usage, and frequency [1]. The reasons behind it are thought-provoking. Based on our practice, we believe that the recombinant growth factors carried by hydrogel scaffolds are easily hydrolysis by some proteases in wounds or washed away by wound exudates. Therefore, to achieve the self-renewability of growth factors is of great clinical significance.

According the international consensus on bioprinting roadmap attached below [2], the application of cells to build blocks in tissue scaffolds is an exciting advancement. More importantly, a creative concept has been proposed and become more popular that cells can not only building blocks for tissue regeneration, but also serve as cell vehicle for delivering therapeutic molecules including growth factors [3]. Our study is just a successful exploration on HUVECs to delivery VEGF 165 in a self-renewable manner. In this round revision, we have added some explanations in the “Introduction part” in lines 132-134, page 7, and supplemented experiments to further confirm the self-renewability of VEGF 165 *in vivo* (**Figure 7F**), thus making our claims more solid.

The bioprinting roadmap.

Additionally, at the cellular and molecular levels, we revealed that the self-renewable VEGF 165 was able to rescue high glucose-induced mitochondrial damages of vascular endothelial cells via Bax/bcl2/cytochrome c/programmed cell death pathways, thus improving angiogenesis and creating satisfactory microenvironments for diabetic wound healing. It deepened the understandings on VEGF 165 for diabetic wound healing. Altogether, although the application of growth factors in wound healing is a

regular practice as you have pointed out, our study has expanded and improved the ways to conduct growth factor therapy based on cell vehicles in regenerative medicine. Thanks again for this thought-provoking comment!

References:

- [1] Marti-Carvajal AJ, Gluud C, Nicola S, et al. Growth factors for treating diabetic foot ulcers. *Cochrane Database Syst Rev.* 2015;2015(10):CD008548.
- [2] Sun W, Starly B, Daly AC, et al. The bioprinting roadmap. *Biofabrication.* 2020;12(2):022002.
- [3] Ding S, O'Banion CP, Welfare JG, Lawrence DS. Cellular Cyborgs: On the Precipice of a Drug Delivery Revolution. *Cell Chem Biol.* 2018;25(6):648-658.

Comment 3. The animal model for wound healing is problematic. Use of STZ to induce diabetes is fine, although the animals are then relatively freshly diabetic. One more commonly uses the db/db mouse for these studies, but still in principle the STZ rat is fine. The main problem is that wound healing in these animals involves a substantial contraction contribution, not just re-epithelialization. One usually then splints the wounds, in which a relatively rigid ring is glued to the skin surrounding the wound, to prevent contraction. This does not seem to have been done here. The results on angiogenesis induction are very clear from the results, but it is not possible to separate the healing effects of contraction from re-epithelialization.

Response 3. Thank you very much for this professional advice! As you suggested, we have added experiments on diabetic wound models with the contraction restricted by silicone rings (**Figure S11**, attached below). Generally, wound healing was postponed compared with the wounds without silicone ring restrictions due to inhibition of wound contraction. Moreover, HUVEC^{vegfl65+}-laden hydrogel group presented the fastest healing speed of diabetic wounds (**Figure S11B-D**). Granulation tissues of diabetic wounds were covered by regenerated epidermal layers in all groups on day 10 (**Figure S11E and F**). The thickness of granulation tissues and epidermal layers (**Figure S11G and H**), and microvascular density (**Figure S11I and J**) were increased after treatment with HUVEC^{vegfl65+}-laden hydrogel. In a word, consistently with the diabetic wound model without silicone ring restrictions, application of HUVEC^{vegfl65+}-laden hydrogel on the wounds restricted by silicone rings expedited epithelial layer to encroach onto the wound bed.

By the way, wound re-epithelialization was mostly resulted from epidermal stem cell motility [1], during which a well-formed granulation tissue in the wound bed is very important for epidermal cell crawling [2]. In this study, we found that HUVEC^{vegfl65+}-laden hydrogel could promote the formation of granulation tissue by increasing thickness, thus providing a native and in situ scaffold for epidermal stem cell motility and wound re-epithelialization. We hope that you are satisfied with our supplemented experiments and explanations and thanks again for this critical comment!

Figure S11. Evaluation of the effect of HUVEC^{vegfl65+}-laden hydrogel on rat diabetic wounds whose contraction is inhibited by a silicone ring. (A) Experimental scheme. ip, intraperitoneal injection. (B) Diabetic wound healing processes were recorded after the different treatments. Inner diameter of the silicone ring: 1 cm. (C) Re-depiction of wound healing processes. (D) Comparison of wound closure rates following the different treatments. n = 10 for days 1, 7, and 10; n = 5 for days 14 and 21. (E) HE analysis revealed varied degrees of granulation tissue formation in the different treatment groups on day 10. (F) CK10 immunofluorescence (IF) staining showed varied degrees of re-epithelization in the different treatment groups on day 10. White arrows: regenerated epithelial layers. (G) Quantitative analysis of granulation tissue thickness in the different treatment groups on day 10. n = 5. (H) Quantitative analysis of regenerated epithelial layer thickness in the different treatment groups on

day 10. n = 5. (I) Representative vascular staining of regenerated granulation tissues in the different treatment groups on day 10. (J) Comparison of vascular density in the different treatment group on day 10. n = 10 because two random visual fields were selected for each of the five samples. The p values in the figure (D) and figure (G, H, and J) were determined by two-way ANOVA and one-way ANOVA, respectively, followed by Tukey's multiple comparisons test, and depicted with asterisks as follows: *, $p < 0.05$; **, $p < 0.01$; ***, $p < 0.001$; ns, not significant.

References:

- [1] Nanba D, Toki F, Asakawa K, et al. EGFR-mediated epidermal stem cell motility drives skin regeneration through COL17A1 proteolysis. *J Cell Biol.* 2021;220(11):e202012073.
- [2] Rousselle P, Montmasson M, Garnier C. Extracellular matrix contribution to skin wound re-epithelialization. *Matrix Biol.* 2019;75-76:12-26.

Finally, on behalf of all the authors, I would like to express my gratitude again for your meticulous review work.

Responses to Reviewer #2:

General impression: I have reviewed the manuscript entitled "Self-renewable VEGF 165 Generated by Click Chemistry-mediated All-peptide Cell Printing Hydrogel Platform for Diabetic Wound Healing via Improving Mitochondrial Function" by Jinjian Huang et al., which describes a novel approach for diabetic wound healing using a self-generating hydrogel platform. Overall, I find this work to be of high quality and the results to be of significant interest for the field of regenerative medicine. However, there are several points that need to be addressed before the manuscript can be considered for publication.

Our response: Thank you very much for your meticulous review of our manuscript and providing valuable suggestions. We sincerely appreciate the time and effort you have dedicated to this process. In this response, we will address each of your comments in order and provide detailed explanations of how we have addressed them.

Comment 1. The authors established a high glucose-induced injury model and determined the appropriate glucose concentration using the CCK-8 assay. However, they did not consider the effect of osmotic pressure changes on cell viability. I suggest the authors conduct additional experiments using mannitol to exclude the effect of osmotic pressure on cell viability.

Response 1. Thank you very much for this advice! As you suggested, we have supplemented an osmotic control (OC) with 5 mM glucose and 35 mM D-mannitol (Figure 6A, attached below). Based on a joint analysis on OD value in 5 mM glucose group, 40 mM glucose group or OC group, it was found that cell viability was not impaired by the increase of osmotic pressure, but affected by the increase in glucose concentration.

Figure 6. (A) HG impairs the viability of HUVECs. OC, osmotic control with 5 mM glucose and 35 mM D-mannitol. $n = 3$. *, $p < 0.05$; ***, $p < 0.001$; ns, not significant.

Comment 2. In Figure 5, the time points for Wound healing and Tube formation assays are not consistent. The authors should clarify why different time points were chosen for these experiments.

Response 2. Thank you very much for this reminding! The time points chosen for wound healing and tube formation assays were based on the cell behavior patterns in

the experiments. Specifically, for the wound healing assay, the migration rate of HUVECs is usually slow; therefore, the duration to observe cells is relatively long, and can be extended to 12 hours or even longer in previous studies [1, 2]. However, for the tube formation assay, HUVECs were seeded on a thin layer of Matrigel rather than directly contacting on bottom of culture plate. This facilitated the spread of cells and formation of tubes in a shorter time. Within 3 hours, we have observed significant tube formation effects after different treatments, which is consistent with the findings in literatures [3, 4]. If over 24 hours, the cells would be aged and dead because of culture on Matrigel. Therefore, we chose different time points to observe therapeutic effects for the two different assays.

References:

- [1] Jonkman JE, Cathcart JA, Xu F, et al. An introduction to the wound healing assay using live-cell microscopy. *Cell Adh Migr.* 2014;8(5):440-451.
- [2] Shen J, Sun Y, Liu X, et al. EGFL6 regulates angiogenesis and osteogenesis in distraction osteogenesis via Wnt/ β -catenin signaling. *Stem Cell Res Ther.* 2021;12(1):415.
- [3] Kelley M, Fierstein S, Purkey L, DeCicco-Skinner K. Endothelial Cell Tube Formation Assay: An In Vitro Model for Angiogenesis. *Methods Mol Biol.* 2022;2475:187-196.
- [4] Gentile MT, Pastorino O, Bifulco M, Colucci-D'Amato L. HUVEC Tube-formation Assay to Evaluate the Impact of Natural Products on Angiogenesis. *J Vis Exp.* 2019;(148):10.3791/58591.

Comment 3. In Figure 6i, the internal reference bands are inconsistent, and the cleaved caspase-3 band is not clear. The expression levels of Cyt-c, Bax, and Bcl2 do not match with the provided internal control β -actin. Please provide new data with appropriate internal controls. I recommend the authors to provide more reliable and comprehensive data.

Response 3. Thank you very much for this comment! As you requested, we have re-performed WB tests and provided new data (**Figure 6I**, attached below) with appropriate internal controls (Cox iv as a mitochondrial loading control; β -actin as a cytosol loading control). The semi-quantitative analysis on bands of interest is shown in **Figure S7**. Uncropped versions of gels for western blot have been deposited in the Source Data file.

Figure 6. (I) Western blot results indicated that the VEGF 165 in the supernatant of HUVEC^{vegfl65+}-laden hydrogel platform alleviated Bax-elicited mitochondrial perforation and casp-3-activated programmed cell death due to mitochondrial leakage.

Comment 4. The authors selected a glucose concentration of 40 mM for the high glucose model, and it is unclear whether the osmotic pressure at this concentration affects cell proliferation. I suggest the authors investigate the effect of osmotic pressure on cell proliferation under 40 mM glucose conditions.

Response 4. Thank you very much for this comment, which is similar with **Comment 1**. As we have explained, we have added osmotic control with 5 mM glucose and 35 mM D-mannitol and confirmed that the change of osmotic pressure would not lead to significant impacts on cell proliferation. In addition, we compared our experimental results with previous publications [1, 2], which turned out to be consistent.

References:

- [1] Niu C, Chen Z, Kim KT, et al. Metformin alleviates hyperglycemia-induced endothelial impairment by downregulating autophagy via the Hedgehog pathway. *Autophagy*. 2019;15(5):843-870.
- [2] Li B, Li H, Dai L, et al. NIK-SIX1 signalling axis regulates high glucose-induced endothelial cell dysfunction and inflammation. *Autoimmunity*. 2022;55(2):86-94.

Comment 5. In Figure 7, the authors compare the wound healing rate at different time points and evaluate blood vessel formation only on day 4 using LASCA analysis. However, in lines 604-605, the authors state that tissue samples were collected on days 3, 7, and 14. It is unclear why they did not do LASCA analysis on days 7 and 14. The authors should provide more detailed information about their experimental design.

Response 5. Thank you very much for this comment! LASCA analysis is a non-invasive and in situ detecting method on wound vascularization, and doesn't need to harvest tissue samples, therefore it is fine for the time point of LASCA analysis to not stay synchronized with the date of tissue harvest. We set the time point of day 4 to perform LASCA analysis because it started transition to proliferative stages of wound healing, and angiogenesis was very active. Moreover, as we have introduced in Materials and methods part (lines 707-708, pages 27-28), we also performed in situ oxygen detection on day 4 using a luminescent oxygen probe, tris(4,7-diphenyl-1,10-phenanthroline) ruthenium (II) dichloride complex. They are paired in situ experiments to investigate the relations of blood supply, tissue oxygenation, and tissue oxidative stress. Therefore, we decided to compare blood supply using LASCA analysis on day 4 after different treatments of simple hydrogel, HUVEC^{vector}-laden hydrogel, and HUVEC^{vegfl65+}-laden hydrogel. To explain our experimental design more clearly, we have revised **Figure 7A** (attached below).

By the way, we have compared vascular density on days 7 and 14 by tissue IF staining of CD 31 and α -SMA. We would like to add LASCA analysis simultaneously, but we sincerely apologize for not providing the data because LASCA equipment was not owned by our lab and not allowed for reservation in summer holiday. However, even without LASCA data on days 7 and 14, we believe that IF staining is a reliable

detecting method for angiogenesis [1, 2], which confirms that HUVEC^{veg165+}-laden hydrogel can promote angiogenesis of diabetic wound healing.

Figure 7. (A) Experimental scheme. ip, intraperitoneal injection; LASCA, laser speckle contrast analysis.

References:

- [1] Teng L, Maqsood M, Zhu M, et al. Exosomes Derived from Human Umbilical Cord Mesenchymal Stem Cells Accelerate Diabetic Wound Healing via Promoting M2 Macrophage Polarization, Angiogenesis, and Collagen Deposition. *Int J Mol Sci.* 2022;23(18):10421.
- [2] Liu L, Zheng CX, Zhao N, et al. Mesenchymal Stem Cell Aggregation-Released Extracellular Vesicles Induce CD31⁺ EMCN⁺ Vessels in Skin Regeneration and Improve Diabetic Wound Healing. *Adv Healthc Mater.* 2023;12(20):e2300019.

Comment 6. The authors explain that collagen remodeling can reflect scar formation from lines 340-349. However, in Figure 7B, the wound is not completely healed on day 14. I suggest the authors provide additional data on the complete wound healing time (such as day17 or day 21) to more fully explain their findings.

Response 6. Thank you very much for this helpful suggestion! As you requested, we have supplemented animal experiments and expanded observation time to day 21 (**Figure 7A and B**) when the wounds in each group almost looked healed. Moreover, the wound tissues were harvested and stained with picosirius red. Collagen I/III ratio on day 21 in different groups was compared in **Figure S12** attached below. The results suggested that the value of collagen I/III ratio in the HUVEC^{veg165+}-laden hydrogel group was still closest to the normal value like the situation on day 14, which implied that the scar risk of the HUVEC^{veg165+}-laden hydrogel group was the lowest.

Figure S12. (A) Representative picosirius red staining images of collagen in the wounds on day 21. i: treated with hydrogel; ii: treated with HUVEC^{vector}-laden hydrogel; iii: treated with HUVEC^{vegfl65+}-laden hydrogel. (B) Quantitative analysis of collagen I/III ratio on day 21. n = 10. **, $p < 0.01$; ***, $p < 0.001$; ns, not significant.

Comment 7. Although Figure S8 provides graphs for blood glucose, body weight, and water consumption, it only shows the results from day 1 and does not provide a dynamic change over the entire 14-day period. The authors should provide data for the entire 14-day period.

Response 7. Thank you very much for this suggestion! We have supplemented the dynamic changes of blood glucose, body weight, and water consumption over the entire experimental process from day -3 to day 21 (Figure S8, attached below). It was found that the rats after i.p. injection of STZ at the dose of 70 mg/kg were suffered from diabetes throughout the entire experimental process by presenting abnormal blood glucose, retarded weight growth, and increased water consumption.

Figure S8. Successful creation of rat diabetic model dependent on intraperitoneal injection of streptozotocin at a dose of 70 mg/kg, which was confirmed by (A) significant elevation of blood glucose; (B) decrease in body weight; (C) increase in water consumption. ns, not significant.

Comment 8. A summary of the reviewer: overall, this manuscript presents a novel approach for diabetic wound healing using a self-generating hydrogel platform. However, the authors need to address the above points to improve the manuscript's clarity, accuracy, and overall scientific rigor. I recommend major revisions to this manuscript.

Response 8. Thank you very much for offering these helpful suggestions above, on which we have taken a full consideration and made corresponding revisions or explanations. We hope that you will be satisfied with our revisions. Thanks again for your time and effects on reviewing our manuscript.

Responses to Reviewer #3:

General impression: Huang L. et al., in their manuscript “Self-renewable VEGF 165 Generated by Click Chemistry-mediated All-peptide Cell Printing Hydrogel Platform for Diabetic Wound Healing via Improving Mitochondrial Function” reported on the preparation of blue light-induced 3D-printable hydrogel loaded with VEGF 165-transfected HUVEC for wound healing. This study demonstrates satisfactory experimental quality and complete identification. However, there exist many issues throughout the whole manuscript that need to be resolved/explained, especially the inappropriate relevance between the materials design and the applications. Experiments and data presented in the work were loosely connected. This reviewer does not believe that the current version of the manuscript provides sufficient novelty and quality for publication in Nature Communications. Here are the major comments.

Our responses: Thank you very much for your careful review of our manuscript and providing valuable suggestions. The main line of this study is to design cell vehicle and cell-printing hydrogel platform for self-renewable VEGF 165 delivery to improve diabetic wound healing. All the data on material design, cell effects, and animal experiments have verified the rationality of the cell-printing hydrogel platform and shown potentials for clinical applications. As you suggested, to highlight this main line, we have made lots of revisions and supplemented necessary experiments. In the following responses, we will address each of your comments in order and provide detailed explanations of how we have addressed them.

Comment 1. The composition of the hydrogel used in this study is highly similar to the previous publication by the authors (International Journal of Biological Macromolecules 142 (2020): 332-344), merely with the addition of extra RGDC to the hydrogel system in this study. However, it was demonstrated in previous work that the γ -PGA-GMA/ γ -PGA-SH hydrogel is sufficient to embed bone marrow-derived MSCs with good viability. With this case, why is it necessary to add the extra RGDC in the present study? Furthermore, if adding the RGDC in the γ -PGA-based hydrogel is insufficient, would the all-peptide hydrogel be necessary? Besides, the authors have previously published HA/ γ -PGA-based hydrogels, while the differences between the hydrogels in this study and the previous hydrogels need to be clarified.

Response 1. Thank you very much for this comment! It can be seen from the author list that this study is an interdisciplinary study. The research team of Dr. Rong Yang and Prof. Bo Chi, who focused on the development of γ -PGA-based biomaterials to mimic extracellular matrix [1-3], contributed to the material design and synthesis. γ -PGA is a type of commercially available polypeptides with high batch stability through microbial fermentation and plays the role of a biocompatible substitute of collagen due to the secondary structure similar to that of natural proteins [4].

Although the mixture of γ -PGA-GMA/ γ -PGA-SH has been proposed by the authors in International Journal of Biological Macromolecules, the gelation time of the simple γ -PGA-GMA/ γ -PGA-SH reactive system was ranged from 7 min to several hours, which was too long to meet the requirement of an appropriate bioink. Moreover,

even the pre-gel solution could be solidified, the least concentration of γ -PGA-GMA and γ -PGA-SH needed to reached 9 wt%. Therefore, it wasted materials. In addition, there is a misunderstanding on the use of RGD. In fact, RGD was added in gel system of the study published in International Journal of Biological Macromolecules (you may check it by referring to the screenshot of the publication where RGD supplementation was mentioned) [5], but it existed in a non-covalent crosslinking form due to a low reactive activity, which led to a significant decrease of RGD's stability.

2.8. Cell culture and encapsulation

Firstly, solutions of γ -PGA-GMA (w/w, 11 wt%) and γ -PGA-SH (w/w, 11 wt%) were prepared by uniformly dissolving the polymers in PBS solution containing RGD polypeptide (10 mmol/L), and then sterilized by filtering via 0.22 μ m microporous membrane. Subsequently, cell suspension with a density of 1×10^6 cells

Screenshot. RGD was added to the gel system in the publication of International Journal of Biological Macromolecules.

In this study, we added LAP, a type of blue light initiator into the pre-gel solution to increase the reactive activity. One-step thiol-ene click reaction of γ -PGA-GMA/ γ -PGA-SH and γ -PGA-GMA/RGDC was accomplished within 20 seconds. RGD existed in a covalent stable form as verified by ^1H NMR (**Figure S1**). The gelation time was significantly reduced to meet the requirement of a DLP printable bioink. The concentration of modified γ -PGA used for gelation was decreased to 5.5 wt%. Altogether, the gel system was improved in terms of gelation time, material saving, and stability of RGD compared with that reported previously and more suitable for DLP printing. A table to summarize the difference is attached below, and the relevant content has been added in lines 169-172, page 8, and lines 178-182, page 9.

Comparison of differences in gel system		
	The gel system published previously [5]	The bioink in this study
Composition	Without photo initiators	With the photo initiator of LAP
Gelation time	>7 min, even for several hours	Within 20 seconds
Working concentration of modified γ-PGA	>9 wt%	4.5-6.0 wt%
Existence form of RGD	non-covalent crosslinking, not stable	covalent crosslinking, more stable
Usage	as an injectable hydrogel	as a cell-printing bioink

References:

- [1] Wang P, Pu Y, Ren Y, et al. Dynamic regulable sodium alginate/poly(γ -glutamic acid) hybrid hydrogels promoted chondrogenic differentiation of stem cells. *Carbohydr Polym.* 2022;275:118692.
- [2] Yang R, Liu X, Ren Y, et al. Injectable adaptive self-healing hyaluronic acid/poly (γ -glutamic acid) hydrogel for cutaneous wound healing. *Acta Biomater.* 2021;127:102-115.
- [3] Xu T, Yang R, Ma X, et al. Bionic Poly(γ -Glutamic Acid) Electrospun Fibrous Scaffolds for Preventing Hypertrophic Scars. *Adv Healthc Mater.* 2019;8(13):e1900123.

[4] Ogunleye A, Bhat A, Irorere VU, Hill D, Williams C, Radecka I. Poly- γ -glutamic acid: production, properties and applications. *Microbiology*. 2015;161(Pt 1):1-17.

[5] Yang R, Wang X, Liu S, et al. Bioinspired poly (γ -glutamic acid) hydrogels for enhanced chondrogenesis of bone marrow-derived mesenchymal stem cells. *Int J Biol Macromol*. 2020;142:332-344.

Comment 2. The modulus of living skin ranges from 10-100 kPa (Nature Reviews Materials 5 (2020): 351-370), while the hydrogel studied in this article has a modulus of 0.1-1 kPa, which may be considered too soft. Would it be better to use a material with a modulus closer to that of skin?

Response 2. Thank you very much for recommending this review paper and raising questions on relation of biomaterial stiffness and skin mechanics. We'd like to discuss this critical issue based on clinical experience, literatures, and experimental data from our lab.

First, from a clinical application perspective, biomaterials required to present similar mechanical strength with native tissues are usually these implanted into load-bearing positions such as cartilages (e.g., meniscus), and bones of four limbs. As for skin wound, it is usually an open injury and doesn't need to tolerate mechanical stress, therefore, there is no strict restrictions on the mechanical property of hydrogels used for wound care as long as hydrogels are compliant with the natural deformation of skin wounds.

Moreover, evidence from basic medical science researches indicated that pulling skin by self-inflating hydrogel tissue expander (namely, Osmed) rather than hydrogel itself was able to promote skin regeneration [1]. Epidermal stem cells underwent renewal due to skin expansion. Single cell RNA-sequencing identified that a population of undifferentiated cells with a stem cell-like signature was proportionately increased after stretching, which exhibited increased activity of transcription factors regulating proliferation, inflammation, and commitment or differentiation. For this basic and fundamental finding, our research group specially proposed a new concept of "wound-contractible hydrogel for skin regeneration, a new insight from mechanobiology", which was published in Matter [2]. We believed that adhesive hydrogels pulling skin by deformation were able to induce skin regeneration. Subsequently, we designed a novel adhesive temperature-sensitive mechanically active hydrogel dressing, based on methacrylic anhydride modified gelatin, N-isopropylacrylamide, and acrylic acid. This hydrogel was able to contract wound and activate MEK/ERK and YAP signaling to regulate wound healing [3]. Thus, we think that the priority should be given to contractive forces on wounds when we designed a mechanically active hydrogel dressing.

However, different from the mechanically active hydrogel, this study was aimed to develop a cell vehicle for self-renewable VEGF 165 delivery based on a cell-printing platform. The storage modulus of Gel 3 used for cell printing was \sim 570 Pa, which was within the storage modulus range of hydrogel reported previously from tens to thousands Pa [4, 5]. Hence, the stiffness of the hydrogel in this study is reasonable and acceptable. In addition, the cell co-culture experiments in **Figure 4D-H** further verified

that the hydrogel can be a suitable cell carrier of HUVECs because the cell presented a satisfied capacity in proliferation.

References:

- [1] Aragona M, Sifrim A, Malfait M, et al. Mechanisms of stretch-mediated skin expansion at single-cell resolution. *Nature*. 2020;584(7820):268-273.
- [2] Jinjian Huang, Xiuwen Wu, Jianan Ren. Wound-contractible hydrogel for skin regeneration, a new insight from mechanobiology. *Matter*. 2021; 4(10): 3091-3094.
- [3] Li Z, Huang J, Jiang Y, et al. Novel Temperature-Sensitive Hydrogel Promotes Wound Healing Through YAP and MEK-Mediated Mechanosensitivity. *Adv Healthc Mater*. 2022;11(23):e2201878.
- [4] Ma C, Liu K, Li Q, et al. Synthetic Extracellular Matrices for 3D Culture of Schwann Cells, Hepatocytes, and HUVECs. *Bioengineering (Basel)*. 2022;9(9):453.
- [5] Liu Y, Zhang Y, An Z, et al. Slide-Ring Structure-Based Double-Network Hydrogel with Enhanced Stretchability and Toughness for 3D-Bio-Printing and Its Potential Application as Artificial Small-Diameter Blood Vessels. *ACS Appl Bio Mater*. 2021;4(12):8597-8606.

Comment 3. The authors indicated that Gel 3 was chosen for evaluating DLP printability due to its improved mechanical properties. What is the basis for selecting from Gels 1, 2, and 3? Are there higher concentration groups, such as Gels 4, 5, and 6?

Response 3. Thank you very much for this suggestion! We have supplemented a group of Gel 4 with the different concentration (6 wt%) of modified polymers (**Table 1**, attached below). Each characterization approach on Gel 1, 2, and 3 was also carried out on Gel 4 including SEM, rheological property, compressive curve, swelling ratio and so on. The new data can be found in **Figure 2** and **Figure S2-5**. Based on the compressive curves of Gel 1-4, it was shown that the compressive strength was gradually increased, but the strain at break was decreased with increase in polymer concentrations. Among all gel formulas, Gel 3 presented the largest fracture energy, as a result of balance in compressive stress and flexibility (**Figure 2G-J**, attached below). Therefore, the formula of Gel 3 was chosen for the evaluation of DLP printability. The above explanations have been added in lines 188-191, page 9, and lines 198-199, page 10.

Samples	PGA-SH	PGA-GMA	RGDC	LAP
Gel 1	4.5 w/v%	4.5 w/v%	0.1 w/v%	0.1 w/v%
Gel 2	5.0 w/v%	5.0 w/v%	0.1 w/v%	0.1 w/v%
Gel 3	5.5 w/v%	5.5 w/v%	0.1 w/v%	0.1 w/v%
Gel 4	6.0 w/v%	6.0 w/v%	0.1 w/v%	0.1 w/v%

Table 1. Hydrogel samples composed of different formulas.

Figure 2. (G) Representative compressive stress-strain curves of different hydrogels. (H) Compressive strength of hydrogels with relatively high polymer concentration (Gel 3 and Gel 4) is increased compared to that of hydrogels with lower polymer concentration (Gel 1 and Gel 2). $n = 3$. (I) Strain at break of hydrogels is decreased with increase in polymer concentration. $n = 3$. (J) Fracture energy of Gel 3 is the highest among all the groups of hydrogels. $n = 3$. *, $p < 0.05$; **, $p < 0.01$; ***, $p < 0.001$; ns, not significant.

Comment 4. A resolution power of 1 mm appears insufficient to guarantee high-precision printing. Could the Pr value, U value, or other printing parameters (On the progress of hydrogel-based 3D printing: Correlating rheological properties with printing behavior. International Journal of Pharmaceuticals, 121506,2022) of all-peptide hydrogels be quantified?

Response 4. Thank you very much for this comment and recommending the review to us. We have read this paper carefully, but find that this review is a summary of extrusion-based printing rather than DLP printing. The two printing technologies are quite different in the materials, printer, and printing mechanism, speed, and resolution [1]. Therefore, the parameters in this review article are not applicable to the bioink in our study. By referring to previous literatures, several parameters affecting the resolution of DLP printing have been reported including bioink category, printing speed, light exposure time, and printer performance such as resolution of LED projector [2, 3]. Because the bioink and printer used in this study were determined, we have comprehensively investigated the remaining influencing factors of printing speed and light exposure time to optimize printing resolution and efficiency (Figure 3).

It is worth noting that in the original manuscript, we have successfully achieved the smallest gap of the comb model at 1 mm. In this round of revision, we adjusted the comb model with a gradient reduction of gaps from 3 mm to 0.25 mm (Figure 3I and J, attached below). In this way, we revealed the smallest resolution to be 0.5 mm because the gap at 0.5 mm was the minimum distance required to separate adjacent comb teeth. Moreover, in order to figure out the printing error range, we designed a microtube array in which the wall thickness of microtube was 0.5 mm. After printing, the wall of microtube was measured as 0.622 mm under confocal microscopy by staining the wall using fluorescein isothiocyanate (FITC). Therefore, the printing error was calculated to be 0.122 mm (Figure 3K, attached below). This finding could interpret the disappearance of comb teeth's gap at the interval of 0.25 mm because the printing error range of adjacent comb teeth was accumulated to nearly 0.25 mm. The

revisions above which attempted to figure out printing resolution and errors have been added in lines 220-232, pages 10-11.

Figure 3. (I) Comb model for the determination of printing resolution. (J) Real printed comb with the printing resolution reaching 0.5 mm because the gap at 0.5 mm was the minimum distance required to separate adjacent comb teeth. White bars = 5 mm. (K) Printing microtube models with the wall thickness at 0.5 mm. The wall thickness of real microtubes was measured to be 0.622 mm, reflecting that the printing error was 0.122 mm. White bar = 5 mm. H, height; W, width; T, thickness.

References:

- [1] Daly AC, Prendergast ME, Hughes AJ, Burdick JA. Bioprinting for the Biologist. *Cell*. 2021;184(1):18-32.
- [2] Fritschen A, Bell AK, Königstein I, Stühn L, Stark RW, Blaeser A. Investigation and comparison of resin materials in transparent DLP-printing for application in cell culture and organs-on-a-chip. *Biomater Sci*. 2022;10(8):1981-1994.
- [3] Mo X, Ouyang L, Xiong Z, Zhang T. Advances in digital light processing of hydrogels. *Biomed Mater*. 2022;17(4):10.1088/1748-605X/ac6b04.

Comment 5. What is the self-renewal ability of the HUVEC $veg165^{+}$ -laden hydrogel and how was it compared with other hydrogels? Meanwhile, what is the release mechanism by which the self-renewable VEGF 165 released from the HUVEC $veg165^{+}$ -laden hydrogel promotes wound healing, cell proliferation, and tube formation of HUVECs *in vitro*? The authors should add more discussion about this point, which the authors highlighted.

Response 5. Thank you very much for this critical comment! We think that self-renewal ability of VEGF 165 was resulted from the cell vehicle of HUVEC $veg165^{+}$ encapsulated in the hydrogel. VEGF 165 is a splice variant of VEGFA that can be secreted extracellularly [1, 2]. Then, the proteins are released out of hydrogel driven by concentration gradients to achieve biological functions [3]. The hydrogel does not produce VEGF 165 directly, but offers a mimetic ECM scaffold for the cell vehicle to generate VEGF 165. This therapeutic approach is an emerging research field, namely cellular cyborgs [4]. To further enhance the production of VEGF 165, we treated the cells with the lentivirus containing the *VEGF 165* transcript, and verified the self-renewable and extracellularly-secretory ability of this protein (**Figure 5A and Figure**

S6C, D). The relevant descriptions have been added in lines 104-106, page 6, and lines 132-134, page 7.

References:

- [1] de Paulis A, Prevete N, Fiorentino I, et al. Expression and functions of the vascular endothelial growth factors and their receptors in human basophils. *J Immunol.* 2006;177(10):7322-7331.
- [2] Feng Q, Zhang C, Lum D, et al. A class of extracellular vesicles from breast cancer cells activates VEGF receptors and tumour angiogenesis. *Nat Commun.* 2017;8:14450.
- [3] Mirjalili F, Mahmoodi M. Controlled release of protein from gelatin/chitosan hydrogel containing platelet-rich fibrin encapsulated in chitosan nanoparticles for accelerated wound healing in an animal model. *Int J Biol Macromol.* 2023;225:588-604.
- [4] Ding S, O'Banion CP, Welfare JG, Lawrence DS. Cellular Cyborgs: On the Precipice of a Drug Delivery Revolution. *Cell Chem Biol.* 2018;25(6):648-658.

Comment 6. What is the mechanism of HUVEC $veg165^+$ -laden hydrogel on re-epithelialization, hair formation, scarless tendency, angiogenesis, and cell proliferation and metabolism of diabetic wounds? The authors only showed significant results but did not give enough discussion between self-renewable VEGF165 and these effects.

Response 6. This is a helpful suggestion enlightening us to reflect on why comprehensive biological effects were changed by HUVEC $veg165^+$ -laden hydrogel. First, it can be concluded that the enhanced and self-renewable VEGF 165 is the reason accounting for these various effects because of the control variable design of experiments. Theoretically, angiogenic process could be directly promoted by VEGF 165, but in fact histological and proteomic data suggested that many effects other than angiogenesis has been produced. To explain potential reasons, we further performed a joint analysis of protein-protein interaction and protein-biological process relation, which indicated that some differentially expressed proteins related to angiogenesis were simultaneously involved in inflammatory response and cell-matrix adhesion (**Figure 9F**, attached below). The inflammatory states and matrix deposition have been reported to regulate wound healing process such as re-epithelialization, scar formation, and tissue oxygenation and metabolism [1, 2]. Therefore, these biological processes were improved other than angiogenesis by use of HUVEC $veg165^+$ -laden hydrogel. The relevant discussions have been added in lines 450-461, pages 18-19.

F

Figure 9. (F) Protein-protein interaction and protein-biological process relation network analysis indicates that different biological processes during wound healing including inflammatory response, angiogenesis, and cell-matrix adhesion are interrelated based on some common differentially expressed proteins.

References:

[1] Rousselle P, Montmasson M, Garnier C. Extracellular matrix contribution to skin wound re-epithelialization. *Matrix Biol.* 2019;75-76:12-26.
 [2] DiPietro LA. Angiogenesis and wound repair: when enough is enough. *J Leukoc Biol.* 2016;100(5):979-984.

Comment 7. The description and the target application (wound dressing or artificial skin) of the hydrogel platform in wound healing experiments *in vivo* is not clear. Has the HUVECvegfl65+-laden hydrogel platform ever been replaced within 14 days? If not, the degradation of the hydrogel is important. The authors should show the status of remaining hydrogels after attaching chronic wounds within 14 days. Also, the viability of VEGF 165-transfected HUVEC in the 3D-printed hydrogel should be specifically demonstrated. If the HUVECvegfl65+ is unable to survive up to 14 days in the hydrogel covering the wounds, the self-renewal ability of VEGF 165 highlighted in this work should be explained in detail.

Response 7. Thank you very much for the reminding! Given the self-renewable ability VEGF 165 by the cell printing platform, we did not replace the hydrogels in the entire animal experiment. The relevant descriptions have been added in lines 685-686, page 27.

Moreover, the *in vivo* degradation tests by calculating remaining weight proportion of hydrogels have been supplemented (**Figure 7E**, attached below).

Figure 7. (E) Hydrogel degradation rate in the three treatment groups calculated by residue weight. n = 5.

To demonstrate the cell viability in hydrogels and the resultant self-renewable capacity by a HUVEC vehicle, we measured the gene transcriptional activity by detecting the green fluorescent protein transcribed by lentivirus (**Supporting Figure**, attached below), and the concentration of VEGF 165 in the hydrogel (**Figure 7F**, attached below). The data indicated that VEGF 165 in the cell-laden hydrogels showed a regenerative manner, especially for the HUVEC^{vegfl65+}-laden hydrogel group in the first four days. After that, the gene transcriptional activities and production of VEGF 165 were decreased possibly due to the lack of nutrients in the hydrogels. The relevant content has been described in lines 344-350, page 15.

Supporting Figure. Gene transcriptional activity of cell-laden hydrogels by detecting the green fluorescent protein transcribed by lentivirus.

Figure 7. (F) Dynamic changes of VEGF 165 concentration in HUVEC^{vector}-laden hydrogels and HUVEC^{veg165+}-laden hydrogels during the *in vivo* treatment process. n = 5.

Comment 8. The wound exudate of chronic wounds is a serious problem. Would the exudate influence the release efficiency or the function of VEGF 165 from the hydrogel? Could this hydrogel system absorb excessive wound exudate? If not, it may cause infection when the wound exudate accumulated.

Response 8. Thank you very much for this critical comment! We completely agree that an appropriate hydrogel should be able to absorb wound exudate and prevent infections [1, 2]. Swelling ratio is a reliable indicator to evaluate hydrogel's ability to absorb wound exudate. In the animal experiments, the swelling ratio of the wet printed hydrogels by formula of Gel 3 was ~260% (**Figure S3**, attached below), which implied that the hydrogels were able to absorb a certain amount of wound exudate. Therefore, we did not observe the accumulation of wound exudate and serious infections such as pus during the animal experiments.

Figure S3. Swelling ratio of Gel 1, Gel 2, Gel 3, and Gel 4. n = 3.

References:

[1] Broussard KC, Powers JG. Wound dressings: selecting the most appropriate type. *Am J Clin Dermatol.* 2013;14(6):449-459.

[2] Ghobril C, Grinstaff MW. The chemistry and engineering of polymeric hydrogel adhesives for wound closure: a tutorial. *Chem Soc Rev.* 2015;44(7):1820-1835.

Comment 9. The authors mentioned that the wounds were followed by treatment with DLP printing of hydrogels. However, the round wound model is not complicated to highlight the necessity of using DLP printing. In Figure 7B, the hydrogels on the wound as shown appear to be bulk hydrogel rather than 3D-printed hydrogel. The reliability and quality of histological figures are not enough. The data seemed doubtful and may mislead the readers.

Response 9. Thank you very much for raising these concerns! We created the diabetic wound in round because this animal modeling method was well-recognized, widely used, and easy to determine the wound area [1, 2]. Meanwhile, to exhibit the DLP printed hydrogels adaptive to irregularly-shaped wounds, we supplemented experiments by creating wounds in different shapes including triangle, rectangle, and heart shaped. Then, hydrogels in the corresponding shapes were designed and printed. The produced hydrogels were well adapted to the wounds (**Figure S9A**, attached below).

Moreover, we'd like to stress that the hydrogels used in the animal study were all DLP printed (**Figure S9B**, attached below). As the DLP printing is layer-by-layer photocuring printing technology, we can print multiple round cell-laden hydrogels at a time. The relevant content has been added in lines 339-341, page 14.

Figure S9. (A) Shape adaptability for irregular wounds by DLP printing of hydrogel. i: creating wounds in different shapes; ii: designing hydrogel shapes; iii: DLP printing of hydrogels; iv: the shape of printed hydrogels was adaptive to the wounds. (B) High throughput production of DLP printed hydrogels. The example showed that four hydrogels were printed simultaneously. i: designing four hydrogels; ii: DLP printing of the designed hydrogels.

References:

[1] Qian Y, Zheng Y, Jin J, et al. Immunoregulation in Diabetic Wound Repair with a Photoenhanced Glycyrrhizic Acid Hydrogel Scaffold. *Adv Mater.* 2022;34(29):e2200521.

[2] Guan Y, Niu H, Liu Z, et al. Sustained oxygenation accelerates diabetic wound healing by promoting epithelialization and angiogenesis and decreasing inflammation. *Sci Adv.* 2021;7(35):eabj0153.

Comment 10. This reviewer strongly recommends that the authors re-clarify the linkages between each part of this study. The current version of the manuscript fails to clearly state convincing reasons for the selection of materials, the necessity of 3D-printed hydrogel in animal studies and the discrepancy in effectiveness, and the doubts about the use of HUVEC $vegfl65+$ -laden hydrogel in wound dressings. The mechanism by which the HUVEC $vegfl65+$ -laden hydrogel may affect mitochondrial function is a key feature in this study, yet it does not seem to be highlighted in the article and is insufficient to bring this study up to the standard of Nature Communications.

Response 10. Thank you very much for this combinatorial comment not only on some specific issues but also the research framework. To make our responses more clearly, we will respond to the issues point by point first, and then introduce what we have done to made the research framework and content understood by potential readers.

Response to each issue as follows:

(1) Regarding biomaterial selection and advantage. We supplemented one more meaning of γ -PGA-based hydrogel from the perspective of all-peptide hydrogel (lines 113-120, page 6). Specifically, compared with traditional all-peptide hydrogel crosslinked of methacrylate gelatin, methacrylate silk, or methacrylate decellularized extracellular matrix which present uncontrollable variability in their composition between batches due to the differences in sources and processing methods, the γ -PGA-based hydrogel with a definite chemical composition can not only serve well as cell scaffolds, but also ensure the results to be more reproducible and stable during applications. Moreover, modifications on γ -PGA based on click chemistry enabled the quick and precise gelation reaction, thus making an appropriate bioink for DLP printing (lines 169-172, page 8).

(2) Regarding necessity of DLP printing for hydrogel. As we have verified in **Figure S9A**, DLP printable hydrogels could adapt to the complex shaped wounds and meet the clinical application requirements.

(3) Regarding treatment effectiveness. As shown in **Figure 7A-D**, rat diabetic wounds were randomly treated with hydrogels, HUVEC $vector$ -laden hydrogels, or HUVEC $vegfl65+$ -laden hydrogels. Consequently, HUVEC $vegfl65+$ -laden hydrogels could lead to a more rapid wound healing compared with the other two treatments. Moreover, in this round of revision, we supplemented a type of modified diabetic wound with the wound contraction restricted by silicone rings (**Figure S11**, attached below). Generally, wound healing was postponed compared with that without silicone ring restrictions due to inhibition of wound contraction. HUVEC $vegfl65+$ -laden hydrogel group presented the fastest healing speed of diabetic wounds (**Figure S11B-D**). Granulation tissues of diabetic wounds were covered by regenerated epidermal layers in all groups on day 10 (**Figure S11E and F**). The thickness of granulation tissues and epidermal layers (**Figure S11G and H**), and

microvascular density (Figure S11I and J) were increased after treatment with HUVEC^{veg165+}-laden hydrogel. In a word, application of HUVEC^{veg165+}-laden hydrogel was able to promote diabetic wound healing regardless of wound contraction, and the treatment effectiveness was definite.

Figure S11. Evaluation of the effect of HUVEC^{veg165+}-laden hydrogel on rat diabetic wounds whose contraction is inhibited by a silicone ring. (A) Experimental scheme. ip, intraperitoneal injection. (B) Diabetic wound healing processes were recorded after the different treatments. Inner diameter of the silicone ring: 1 cm. (C) Re-depiction of wound healing processes. (D) Comparison of wound closure rates following the different treatments. n = 10 for days 1, 7, and 10; n = 5 for days 14 and 21. (E) HE analysis revealed varied degrees of granulation tissue formation in the different treatment groups on day 10. (F) CK10

immunofluorescence (IF) staining showed varied degrees of re-epithelization in the different treatment groups on day 10. White arrows: regenerated epithelial layers. (G) Quantitative analysis of granulation tissue thickness in the different treatment groups on day 10. $n = 5$. (H) Quantitative analysis of regenerated epithelial layer thickness in the different treatment groups on day 10. $n = 5$. (I) Representative vascular staining of regenerated granulation tissues in the different treatment groups on day 10. (J) Comparison of vascular density in the different treatment group on day 10. $n = 10$ because two random visual fields were selected for each of the five samples. The p values in the figure (D) and figure (G, H, and J) were determined by two-way ANOVA and one-way ANOVA, respectively, followed by Tukey's multiple comparisons test, and depicted with asterisks as follows: *, $p < 0.05$; **, $p < 0.01$; ***, $p < 0.001$; ns, not significant.

- (4) **Regarding therapeutic mechanisms.** Judged by the control variable design of experiments, the reason that promotes wound healing is mainly due to the enhanced and self-renewable VEGF 165. **Figures 5 and 6** has fully demonstrated the therapeutic mechanisms that relies on protective functions on vascular endothelial cells. **Figure 5C-H** verifies that HUVEC^{veg165+}-laden hydrogel promotes cell proliferation, migration, and tube formation. **Figure 6** reveals the molecular mechanisms of VEGF 165 released from cell-laden hydrogel by rescuing Bax-mediated mitochondrial perforation and resultant programmed cell death. The improved angiogenesis is able to improve wound healing, otherwise, it will lead to tissue hypoxia, oxidative stress, and inflammation. Such a molecular mechanism has deepened our understanding on VEGF 165 for wound healing and is depicted as a part of **Graphic Abstract** (attached below).

Graphic Abstract. HUVEC^{veg165+}-laden hydrogel releases VEGF 165 in a sustainable manner to rescue Bax-mediated mitochondrial perforation and resultant programmed cell death of vascular endothelial cells. The improved angiogenesis can improve wound healing; otherwise, it will lead to tissue hypoxia, oxidative stress, and inflammation of other repair-related cells.

To further verified the molecular mechanism, we have supplemented new evidence by preparing single cell suspension (~600,000 cells in total) dissolved from wound tissues of three rats in different groups. 100,000 cells were isolated for cell staining of CD31-FITC and MitoSOX Red in turn, and then detected with flow cytometry. It was found that the proportion of vascular endothelial cells with mitochondrial

damage in the HUVEC^{vegfl65+}-laden hydrogel group was 23.2%, which was much lower than the values of the simple hydrogel group (44.1%) and HUVEC^{vector}-laden hydrogel group (41.8%) (**Figure 7G**, attached below). It provides reliable data confirming that the enhanced and self-renewable VEGF 165 can protect vascular endothelial cells from high glucose-induced cell damage. This related content has been described in lines 351-358, page 15.

Figure 7. (G) Flow cytometry analysis of single cells lysed from wounds of three rats in each group indicated that the proportion of mitochondrial oxidative stress damage in vascular endothelial cells was 23.2% in the HUVEC^{vegfl65+}-laden hydrogel group, which was significantly lower than that of the hydrogel group (44.2%) and HUVEC^{vector}-laden hydrogel group (41.8%). MitoSOX red marked mitochondrial oxidative stress damage; CD 31 marked vascular endothelial cells.

Defining the research framework more clearly as follows:

(5) Regarding the research framework. As a biomaterial-and-tissue engineering-based interdisciplinary study, we focused on the main line of treating diabetic wounds with feasible and effective cell-printing platform, and elaborated the therapeutic mechanisms. Each part of experiments including biomaterial synthesis, genetical medication of cells, molecular regulations, and effectiveness of *in vitro* and *in vivo* treatment altogether served for this main line and was inseparable. Therefore, to highlight the relations of each part, we have drawn a summary diagram (**Figure 10G**, attached below) to make the potential readers understand this study more clearly and easily.

Figure 10. (G) Therapeutic mechanism of wound healing by HUVEC^{vegf165+}-laden hydrogel. VEGF 165 was continuously released from the HUVEC^{vegf165+}-laden hydrogel to rescue HG-induced vascular endothelial cell death by inhibiting mitochondrial oxidative stress, thus improving tissue angiogenesis and oxygenation, and creating an eligible microenvironment for diabetic wound healing.

Finally, on behalf of all the authors, I would like to express my gratitude again for your meticulous review work.

Responses to Reviewer #4:

General impression: In this study, Huang and colleagues employ a method of fabricating a 3D printed all-peptide hydrogel platform that aids the wound healing process and promotes angiogenesis in the context of diabetes. With the complexity of the disease etiology and the increased rate of diabetic individuals in developing foot ulcers and prolonged infection; diabetes reduces the ability of the skin to repair itself thus dysregulating the wound healing process. The manuscript clearly states this rational. That said, while the results are noteworthy, the below comments and considerations would enhance the quality and impact of this article.

Our response: We really appreciate your time and efforts on reviewing our manuscript, and thank you very much for your recognition on this research. A point-to-point response to your comments are attached below for your evaluation.

Comment 1. The use of biomaterials such as hydrogels as drug carriers for skin regeneration and repair advanced was applied 20 years ago, so the originality on the use of hydrogels is moderate. However, the concept of fabricating a biocompatible and biodegradable delivery system that can easily adapt to the wound shape and release the exact amount of the required growth factors to support the wound healing process is novel as it hasn't been considered in the literature. In addition to looking at the mechanistic work of this paper, the authors have presented the reduction in the inflammatory state of the wound and enhanced mitochondria function, which support their finding and hypothesis.

Response 1. Thank you again for your appreciation of our research! In this round revision, we further added flow cytometry experiments on single cell suspensions dissolved from wound tissues followed by different treatments (**Figure 7G**), which verified the *in vivo* mitochondrial protective functions of HUVEC^{veg165+}-laden hydrogel on vascular endothelial cells, thus making the claims of this study more solid. In addition, a joint analysis of protein-protein interaction and protein-biological process relation was added in **Figure 9F**, which showed that some differentially expressed proteins related to angiogenesis were simultaneously involved in inflammatory response and cell-matrix adhesion. It accounted for not only neovascularization, but also the other extensive biological functions such as inflammatory regulation and ECM remodeling that were achieved by the enhanced and self-renewable VEGF 165 from HUVEC^{veg165+}-laden hydrogel. We believe that these revisions can improve the quality of this manuscript and meet your expectations.

Comment 2. To make the paper more appealing to a broader audience, the definitions on click reaction (line 116) and digital light processing technique should be mentioned in the introduction.

Response 2. Thank you very much for your advice! We have supplemented the definition of click reaction in lines 125-128, pages 6-7 as follows: "Click reaction refers to an approach to develop a set of fast, highly reliable, yield, and selective reactions for the rapid synthesis of useful new compounds and combinatorial libraries through heteroatom links (C-X-C) [1]". Moreover, digital light processing technique is defined

in lines 110-113, page 6 as follows: “digital light processing printing is a high-resolution fast-speed additive manufacturing technology that forms desirable 3D structures through photopolymerization reaction [2]”.

References:

- [1] Kolb HC, Finn MG, Sharpless KB. Click Chemistry: Diverse Chemical Function from a Few Good Reactions. *Angewandte Chemie International Edition*. 2001;40(11):2004-2021.
- [2] Cheng J, Wang R, Sun Z, et al. Centrifugal multimaterial 3D printing of multifunctional heterogeneous objects. *Nat Commun*. 2022;13(1):7931.

Comment 3. In section 2.2, the DLP variable parameters light exposure and layer thickness were considered, are there other parameters that would affect the scaffold properties such as printing speed, polymers flowrate and ambient parameters?

Response 3. Thank you very much for this advice! In this study, the DLP printer (ANYCUBIC Photon Ultra) is commercially manufactured. The slicing software provided by ANYCUBIC is user-friendly and can be accessed at <https://cn.anycubic.com/list/389.html>. In the slicer, light exposure time and layer thickness were two main variable parameters for users to improve the printing quality; therefore we explored how these two adjustable parameters affected the resolution of printed hydrogels in this study. The parameter of printing speed that you mentioned is just controlled by light exposure time per layer and layer thickness, hence we do not need to study it specifically. In addition, unlike extrusion-based printing technologies, the pre-gel solution is in a state of rest in the bioink reservoir, so it doesn't need to study the flowrate on printing quality in the DLP printing process. Moreover, ambient parameters such as temperature and humidity are not allowed to change by ANYCUBIC DLP printer. Notably, the printer is equipped with a plastic light shield that can maintain the ambient parameters stable during printing. Apart from the parameters mentioned above, the strength and accuracy of LED light sources were important factors determining the printing resolution as reported by literatures [1, 2], but the parameters are not allowed to adjust for a specific printer as well. Altogether, based on the commercial and user-friendly ANYCUBIC DLP printer, we have performed a complete investigation on the parameters of light exposure time and layer thickness to improve the printing resolution for our newly-developed printable hydrogel platform. Thanks again for this constructive suggestion!

References:

- [1] Quan H, Zhang T, Xu H, Luo S, Nie J, Zhu X. Photo-curing 3D printing technique and its challenges. *Bioact Mater*. 2020;5(1):110-115.
- [2] Li Y, Mao Q, Yin J, et al. Theoretical prediction and experimental validation of the digital light processing (DLP) working curve for photocurable materials. *Additive Manufacturing*. 2020.101716.

Comment 4. To strengthen the paper concept, it is recommended to expand in the discussion section to include recent publications on this topic, in addition to explaining the clinical effectiveness of the hydrogel platform for wound healing. Below are some

suggestions:

<https://link.springer.com/article/10.1007/s12274-022-4192-y>

<https://www.sciencedirect.com/science/article/abs/pii/S1742706122003233>

<https://www.sciencedirect.com/science/article/abs/pii/S1001841722007100>

<https://www.frontiersin.org/articles/10.3389/fbioe.2019.00342/full>

Response 4. Thank you very much for providing the research papers regarding hydrogels to treat diabetic wounds. We have read them carefully. The hydrogels reported in these papers are well designed based on the characteristics of diabetes wounds. Therefore, we have cited them properly as shown in References 31-34 (lines 863-870, page 33) in our manuscript to strengthen the paper concept.

Comment 5. In conclusion, the manuscript fits within the journal theme and is recommended from publication once the above points are considered.

Response 5. Thanks again for your recognition on this study. We hope that you can be satisfied with our modifications.

REVIEWERS' COMMENTS

Reviewer #1 (Remarks to the Author):

The revisions to the text to more set the work in context of the literature and the addition of the splinted wound study in Figure S11 have addressed my concerns. I have no further comments regarding this revision.

Reviewer #2 (Remarks to the Author):

The authors have addressed all of my concerns. The paper is significantly strengthened by new experiments, additional data analysis and text clarification. I believe the work is now in a format fit for publishing in the journal and is ready to reach the wider scientific community.

Xuqiang Nie, Ph.D., Prof.

Reviewer #3 (Remarks to the Author):

The authors have tried to address the comments. However, for response to Comment 4, a resolution of 0.625 mm is still poor for cell printing. Besides presenting the data, the authors should discuss the possibility of enhancing the resolution or the possibility of using the current hydrogel in the other 3D cell printing methods.

In the meantime, I have also read the response to reviewer #4. I think the authors have addressed the comments of this reviewer in detail.

Responses to Reviewers:

To Reviewer #1:

Comment 1: The revisions to the text to more set the work in context of the literature and the addition of the splinted wound study in Figure S11 have addressed my concerns. I have no further comments regarding this revision.

Response 1: We really appreciate your time and efforts on reviewing our responses, and thank you very much for your recognition on this revision. Our manuscript has been improved a lot by your suggestions.

To Reviewer #2:

Comment 1: The authors have addressed all of my concerns. The paper is significantly strengthened by new experiments, additional data analysis and text clarification. I believe the work is now in a format fit for publishing in the journal and is ready to reach the wider scientific community.

Response 1: Thank you very much for your meticulous review of our manuscript and providing valuable suggestions. We sincerely appreciate the time and effort you have dedicated to this peer review process. Our manuscript has been improved a lot by your suggestions.

To Reviewer #3:

Comment 1: The authors have tried to address the comments. However, for response to Comment 4, a resolution of 0.625 mm is still poor for cell printing. Besides presenting the data, the authors should discuss the possibility of enhancing the resolution or the possibility of using the current hydrogel in the other 3D cell printing methods. In the meantime, I have also read the response to reviewer #4. I think the authors have addressed the comments of this reviewer in detail.

Response 1: Thank you very much for your meticulous review of our manuscript and providing valuable suggestions. Meanwhile, we sincerely appreciate your assistance on evaluating our responses to Reviewer #4.

Regarding the printing resolution, we would like to clarify that the resolution of DLP printing for the bioink was defined as 0.5 mm (**NOT** 0.625 mm) according to the observation on the gaps of the printed comb model (**Figure 3I and J**, attached below), and the printing error was measured to be 0.122 mm (**Figure 3K**, attached below). The data of printing error could interpret the disappearance of comb teeth's gap at the interval of 0.25 mm because the printing error range of adjacent comb teeth was accumulated to nearly 0.25 mm.

Moreover, as requested, we have searched literature about the strategies to improve the printing resolution, which revealed that apart from the optimization of printing parameters, the DLP printing resolution could be further improved by increasing the light source resolution or adding specific biomacromolecule additives to inhibit the free

radical polymerization at unexposed areas during photoinitiation [1-2]. However, as this point was not the main line of this study, we preferred discussing this topic in our near-future 3D printing-specific studies rather than in this study that focused on diabetic wound healing by a new DLP printing cell-laden platform. The current resolution of the printed hydrogel was enough for diabetic wound healing applications; therefore, we wish to gain your understanding.

Finally, we would like to express our gratitude again for your careful peer review work!

Figure 3. (I) Comb model for the determination of printing resolution. (J) Real printed comb with the printing resolution reaching 0.5 mm because the gap at 0.5 mm was the minimum distance required to separate adjacent comb teeth. White bars = 5 mm. (K) Printing microtube models with the wall thickness at 0.5 mm. The wall thickness of real microtubes was measured to be 0.622 mm, reflecting that the printing error was 0.122 mm. White bar = 5 mm. H, height; W, width; T, thickness.

References:

- [1] He X, Cheng J, Sun Z, et al. A volatile microemulsion method of preparing water-soluble photo-absorbers for 3D printing of high-resolution, high-water-content hydrogel structures. *Soft Matter*. 2023;19(20):3700-3710.
- [2] Ma Y, Wei W, Gong L, et al. Biomacromolecule-based agent for high-precision light-based 3D hydrogel bioprinting. *Cell Reports Physical Science*. 2022; 3(8): 100985.